# The shared genetic architecture and evolution of human language and musical rhythm

Gökberk Alagöz [1] ✉, Else Eising [1], Yasmina Mekki[2,3], Giacomo Bignardi [1,4], Pierre Fontanillas [5], 23andMe Research Team*, Michel G. Nivard[6,7], Michelle Luciano [8], Nancy J. Cox[3], Simon E. Fisher [1,9,12] ✉ & Reyna L. Gordon [2,3,10,11,12] ✉

This study aimed to test theoretical predictions over biological underpinnings of previously documented phenotypic correlations between human language-related and musical rhythm traits. Here, after identifying significant genetic correlations between rhythm, dyslexia and various language-related traits, we adapted multivariate methods to capture genetic signals common to genome-wide association studies of rhythm ($N = 606{,}825$) and dyslexia ($N = 1{,}138{,}870$). The results revealed 16 pleiotropic loci ($P < 5 \times 10^{-8}$) jointly associated with rhythm impairment and dyslexia, and intricate shared genetic and neurobiological architectures. The joint genetic signal was enriched for foetal and adult brain cell-specific regulatory regions, highlighting complex cellular composition in their shared underpinnings. Local genetic correlation with a key white matter tract (the left superior longitudinal fasciculus-I) substantiated hypotheses about auditory–motor connectivity as a genetically influenced, evolutionarily relevant neural endophenotype common to rhythm and language processing. Overall, we provide empirical evidence of multiple aspects of shared biology linking language and musical rhythm, contributing novel insight into the evolutionary relationships between human musicality and linguistic communication traits.

The human brain has intricate neural circuitry to process complex communicative signals and behaviours, including speech and music, and the extent of biological overlap between these facets is an important question for the field of neurobiology. Individual differences in rhythm-related skills are correlated with variability in language-related skills, implicating potentially shared underlying neural and genetic architectures[1]. Previous research on the relationship between rhythm and language-related skills used a wide range of task-based tests to measure aspects of rhythm (for example, beat synchronization, rhythm perception and production, and metrical perception) and spoken and written language abilities (for example, word recognition, spelling and phonological awareness). For instance, rhythm perception skills can be measured by quantifying participants' ability to discriminate differences in durations of adjacent tones in melodies, whereas beat synchronization skills can be quantified on the basis of participants' success in tapping along to a metronome[2] or to the beat (pulse) of musical excerpts[3]. Similarly, language-related skills, such as word reading ability, can be measured as the ability to sound out words quickly and accurately in a limited amount of time[4], and spelling skills can be measured by testing participants' abilities to correctly spell out a number of words read aloud by the tester[5]. Importantly, even though language and rhythm measurement tasks involve different signals and stimuli and

capture different skills, studies of individual differences often show phenotypic correlations between the different traits[6]. Nayak et al.[1] compiled information from 25 studies that identified significant positive phenotypic correlations between rhythm and language-related skills, synthesizing findings on a total of 397 children and 606 adults. Consistent associations have been found between rhythm perception, beat synchronization and language-related skills including speech perception, word reading and grammatical skills at various phases in the lifespan from pre-school age through to later adulthood. Despite these lines of evidence showing phenotypic associations between non-linguistic rhythmic processing and language skills, empirical evidence at the intersection between the neurobiological, genetic and evolutionary grounds of these traits remain to be discovered.

Various theoretical frameworks[7–9], such as the revised vocal learning hypothesis[10], provide overarching perspectives on how rhythm and multiple facets of human communication might relate in a neurodevelopmental and evolutionary context. According to the revised vocal learning hypothesis, human vocal learning ability is a pre-adaptation for predictive and tempo-flexible beat synchronization, and beat processing and vocal learning rely on overlapping neural circuits. This view is in line with neural reuse theories, such as neuronal recycling[11] and massive redeployment hypotheses[12], which suggest overlapping neurobiological circuits for language- and rhythm-related skills. Neural reuse theories claim that cultural innovations, such as reading, invade evolutionarily older brain substrates via the reallocation of an existing neural circuit to a new behaviour. Some argue that cognitive systems, such as language and music, are better understood as different uses of similar information processing mechanisms[13], yet the genetic and evolutionary bases of putative shared neural circuits are largely unknown.

To address prominent theories on the evolution of language development and musical rhythm in humans[10], evidence so far has been taken largely from psychology, neuroscience and cross-species comparisons rather than genetics[14,15]. We believe that identifying potential shared genetic architecture between language-related disorders and musical rhythm abilities, and probing the evolutionary past of the implicated genomic regions, can help reveal the neural and biological characteristics of our species that made rhythm and language an asset to human development and behaviour. Importantly, individuals with rhythm impairment have been suggested to show higher predisposition to language-related difficulties, such as developmental language disorder and dyslexia (atypical rhythm risk hypothesis, ARRH)[16]. Given that disorders of language and reading can have long-term health impacts, identifying genetic factors that they share with rhythm impairment may enhance future possibilities for diagnoses and treatment. Moreover, basic science concerning the biological substrates of these fundamental human traits will be informed by new approaches to their potentially shared genetic architecture.

Our work built on two recent genome-wide association studies (GWAS) that represent by far the most well-powered genetic investigations of rhythm-/language-relevant traits so far, one for musical rhythm (beat synchronization, hereafter referred to as rhythm; 'can you clap in time with a musical beat?' $N_{cases}$(yes) = 555,660, $N_{controls}$(no) = 51,165)[17] and the other for dyslexia (developmental reading/spelling difficulties; 'have you been diagnosed with dyslexia?' $N_{cases}$(yes) = 51,800, $N_{controls}$(no) = 1,087,070)[18], both performed on a 23andMe Inc. research cohort in individuals of European ancestry and both classified as binary traits. We used the dyslexia GWAS as a proxy for the genetic underpinnings of language- and reading-related aspects of human communication, as dyslexia often co-occurs with a number of speech/language disorders[19–22]. Beat synchronization GWAS was used as a proxy for musical rhythm skills, as beat perception and synchronization are considered to be important features of musical experiences in present-day humans[23,24]. We applied a three-stage analytic pipeline to investigate shared genetics and biology: (1) genome-wide genetic correlations between rhythm and dyslexia (as well as other language-related traits)

using linkage disequilibrium score regression (LDSC)[25], (2) multivariate GWAS (mvGWAS) of rhythm impairment and dyslexia using genomic structural equation modelling (SEM)[26] and (3) post-mvGWAS analyses of the shared genomic infrastructure as windows into its evolution and biology (Fig. 1a).

## Results

In the first stage, we estimated genetic correlations between rhythm and dyslexia, as well as quantitative measures of language or reading performance[27], educational traits[28] and brain–language-related endophenotypes[29,30] by using LDSC[25]. We found moderate but significant genetic correlations between rhythm and dyslexia (magnitude of the genetic correlation ($r_g$) (s.e.m.) = −0.28 (0.02), $P_{FDR} = 2.05 \times 10^{-31}$), five quantitative language or reading measures, three educational traits and two language-relevant neuroimaging endophenotypes (Fig. 1b and Supplementary Table 1). In contrast there were negligible and non-significant genetic correlations with non-verbal intelligence quotient (IQ) ($r_g$ (s.e.m.) = −0.004 (0.047), $P_{FDR} = 0.94$) and overall school performance ($r_g$ (s.e.m.) = −0.066 (0.040), $P_{FDR} = 0.11$) (Fig. 1b and Supplementary Table 1). Thus, rhythm is genetically correlated not only with dyslexia, but also with multiple language-related phenotypes including word and non-word reading, non-word repetition, phoneme awareness, having better language skills than mathematics, and language resting-state functional connectivity ($|r_g|$ median of 0.184 and range of 0.004–0.376), providing empirical genetic evidence for the ARRH. The absence of significant genetic correlations between rhythm and cognitive traits, such as non-verbal IQ and overall school performance, provides evidence that genetic sharing between rhythm and dyslexia is not driven by general cognition. These results represent the first direct empirical support for a shared genetic architecture underlying previously observed phenotypic correlations between rhythm and language-related traits[1], such as dyslexia (Pearson correlation of −0.04, 95% confidence interval (CI) −0.05 to −0.04, $t = -25.96$, d.f. of 363,285, $P < 2.2 \times 10^{-16}$).

Given that dyslexia is a neurodevelopmental disorder with effects particularly apparent in the written language domain (evident from reading and/or spelling difficulties)[18] and that other work has shown rhythm impairments associated with dyslexia[19–22], we expect it to be genetically and phenotypically linked to impairment in rhythm (hereafter referred to as rhythm impairment) rather than rhythm ability (this expectation is supported by the negative sign of the genetic correlation observed in the first stage of our pipeline above). Thus, we reversed the effect directions in the binary rhythm GWAS summary statistics to align genetic effect directions for rhythm and reading impairments. We then performed a mvGWAS on the rhythm impairment and dyslexia GWAS to probe the validity of the ARRH at the genetic level, using a bivariate extension of genomic SEM[26] that we developed (Methods). This allowed us to tease apart the genetic effects shared between rhythm impairment and dyslexia from those that are unique to each. We specified a measurement model with a shared genetic factor ($F_{gRI-D}$, where RI-D stands for rhythm impairment-dyslexia), which recaptured the genetic correlation between two traits ($\sigma^2 F_{gRI-D}$ (s.e.m.) = 0.28 (0.03)). Similar to Grotzinger et al.[31], we then applied both the common pathway model (CPM), which regresses single nucleotide polymorphisms (SNPs) from $F_{gRI-D}$ (Supplementary Fig. 1), and the independent pathway model (IPM), which regresses SNPs directly onto the genetic components of the two traits (Supplementary Fig. 1). Thus, we were able to obtain a quantitative per-SNP score quantifying the extent to which any given SNP influences rhythm impairment or dyslexia independent from $F_{gRI-D}$, that is, the bivariate genetic heterogeneity ($Q_b$).

Our mvGWAS analysis with the CPM resulted in a new set of summary statistics representing the genetic overlap between rhythm impairment and dyslexia, and identified 18 genome-wide significant ($P < 5 \times 10^{-8}$) loci associated with $F_{gRI-D}$ (Fig. 2a and Supplementary Table 2) after genomic control (GC) correction (Supplementary Fig. 2).

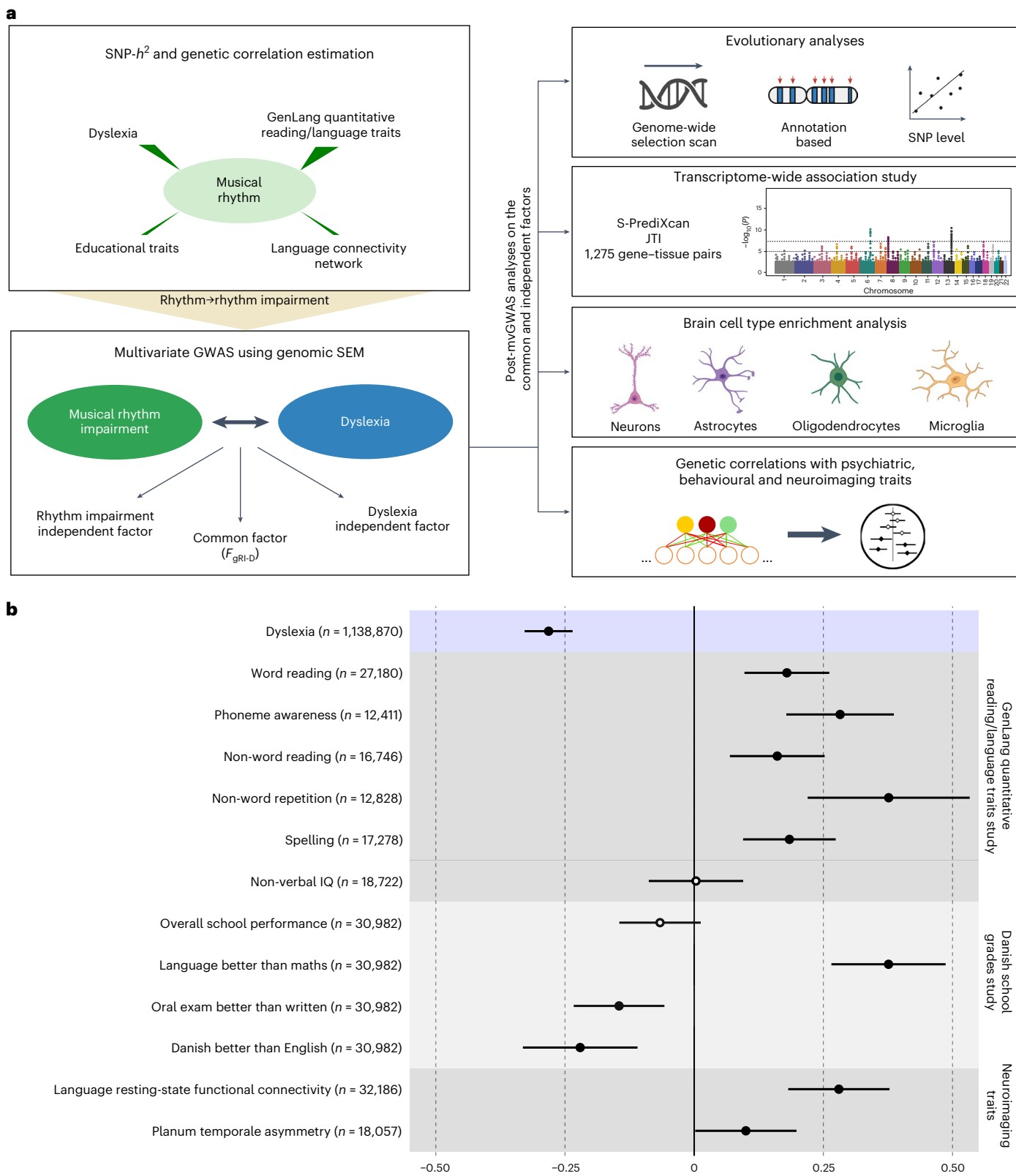

**Fig. 1 | Study design and genetic correlations between rhythm and language or reading-related traits. a**, The flow chart shows the analyses performed in our study. SNP-heritability (SNP-$h^2$) and genetic correlations were estimated using LDSC. The effect directions in the rhythm GWAS summary statistics were flipped to obtain a proxy to probe rhythm impairment. Genomic SEM was used to identify common and independent genetic factors of rhythm impairment and dyslexia. For post-mvGWAS analyses, we adopted various methods including LDSC partitioned heritability, GCTB SBayesS, LAVA and manual SNP lookups.

**b**, The genetic correlation analysis results between musical rhythm and a set of language- and reading-related traits. The $x$ axis indicates the magnitude of the genetic correlation ($r_g$), with error bars extending one s.e.m. below or above the correlation estimate. The $y$ axis shows 13 language or reading-related, cognitive, educational and neuroimaging traits. GWAS sample sizes ($N$) are reported next to the trait names. $P$ values were estimated using a two-sided test and were FDR corrected for 13 tests ($P_{FDR} < 0.05$). Significant genetic correlations ($P_{FDR} < 0.05$) are indicated by full circles. Panel **a** created with BioRender.com.

We estimated the SNP heritability of $F_{gRI-D}$ as 13% (s.e.m. of 0.005) by LDSC[25]. The strongest mvGWAS signal came from the SNP rs28576629 ($P = 3.79 \times 10^{-14}$) on chromosome 3 (Fig. 2a), an intronic variant in *PPP2R3A*, a gene encoding a regulatory subunit of protein phosphatase 2 (ref. 32). We validated the genomic SEM CPM results using two additional mvGWAS methods: (1) *N*-weighted genome-wide association meta-analysis (GWAMA)[33] and (2) cross-phenotype association analysis (CPASSOC)[34]. Both methods captured highly similar genomic architectures to the one captured by the CPM (Supplementary Fig. 3), confirming that the shared genetics of rhythm impairment and dyslexia could be identified consistently regardless of analytic tool. The IPM resulted in two new sets of summary statistics capturing the genetic factors of rhythm impairment and dyslexia that are independent from $F_{gRI-D}$, so-called independent factors (Supplementary Fig. 4). We used the IPM results to obtain $Q_b$ and mapped the per-SNP $Q_b$ scores onto CPM mvGWAS results to dissociate the homogeneous (hereafter referred to as pleiotropic) signals from the signals driven by a single GWAS (Fig. 2a). We identified 27 significant genome-wide ($P < 5 \times 10^{-8}$) heterogeneous loci in the $Q_b$ results (Fig. 2a and Supplementary Table 3), and two of these loci are co-localized with two CPM signals on chromosome 20 (30,690,943–31,189,993 and 47,514,881–47,821,129), which are mvGWAS signals that are driven by the dyslexia GWAS (Fig. 2a). Thus, our analysis revealed two distinct patterns for CPM mvGWAS hit loci: 16 highly homogeneous (putatively pleiotropic) and two heterogeneous loci indicating different levels of GWAS significance, effect sizes and/or opposite effect directions for these two loci in the rhythm impairment and dyslexia GWASs (Fig. 2b shows the representative loci of each type).

Next, we performed a transcriptome-wide association study (TWAS) using $F_{gRI-D}$ summary statistics, and whole-blood and 13 GTEx brain tissue phenotype weights[35,36] with S-PrediXcan[37] (Supplementary Table 4). Our TWAS analysis identified 1,275 significant ($P_{FDR} < 0.05$) gene–tissue pairs and 315 significant ($P_{FDR} < 0.05$) unique genes associated with $F_{gRI-D}$ after false discovery rate (FDR) correction (Fig. 3a and Supplementary Table 5). Some of the top significant gene–tissue pairs associated with $F_{gRI-D}$ are AC072039.2 expression in brain nucleus (*Z*-score of −7.74, $P_{FDR} = 1.17 \times 10^{-9}$), *PPP2R3A* expression in cerebellum (*Z*-score of 7.49, $P_{FDR} = 2.43 \times 10^{-9}$) and putamen (*Z*-score of 7.47, $P_{FDR} = 2.43 \times 10^{-9}$) and *FOXO3* expression in anterior cingulate cortex (*Z*-score of 6.07, $P_{FDR} = 1.15 \times 10^{-5}$) (Fig. 3a). Functional enrichment analysis of the significant ($P_{FDR} < 0.05$) TWAS genes using PANTHER[38] did not identify any significant enrichments in Gene Ontology (GO) and PANTHER GO-Slim terms[38–41] after accounting for multiple testing (Supplementary Tables 6–11). Overall, our S-PrediXcan analysis highlighted 315 unique genes linked to $F_{gRI-D}$, including significant gene–tissue pairs (such as *FOXO3* expression in the anterior cingulate cortex and *PPP2R3A* expression in the putamen) involving brain regions with known relevance for music processing[42,43].

To investigate the neurobiology of genetic variation shared between rhythm impairment and dyslexia at cell type resolution, we performed LDSC partitioned heritability analysis[44] using cell type-specific regulatory region annotations of neurons, microglia, astrocytes and oligodendrocytes[45]. We found robust significant SNP

heritability enrichments in the promoters of neurons (enrichment (SEM) of 8.14 (1.55), $P_{FDR} = 3.38 \times 10^{-5}$), oligodendrocytes (enrichment (SEM) of 7.98 (1.53), $P_{FDR} = 3.38 \times 10^{-5}$), astrocytes (enrichment (SEM) of 7.72 (1.59), $P_{FDR} = 1.1 \times 10^{-4}$) and microglia (enrichment (SEM) of 4.47 (1.63), $P_{FDR} = 0.04$), as well as enhancers of neurons (enrichment (SEM) of 4.43 (0.35), $P_{FDR} = 7.96 \times 10^{-18}$) and astrocytes (enrichment (SEM) of 2.73 (0.58), $P_{FDR} = 4.35 \times 10^{-3}$) (Fig. 3b and Supplementary Table 12). Consistent with the original rhythm and dyslexia GWAS reports[17,18], $F_{gRI-D}$ relates to brain structure in part by common effects at regulatory regions within multiple cell types, including neuronal and various non-neuronal cells, such as oligodendrocytes. This may suggest that the $F_{gRI-D}$ might impact myelination and white matter connectivity patterns that could potentially instantiate neural overlap between rhythm and reading-related aspects of language[1,10,46].

To test the validity of links between rhythm impairment and dyslexia risk and proneness to certain neuropsychiatric disorders proposed by the ARRH, we moved on to investigate relationships of $F_{gRI-D}$ with psychiatric, neurological and behavioural traits, examining patterns of genetic correlations with common and independent factors in more detail. First, we curated 88 sets of GWAS summary statistics including traits that were significantly genetically correlated either with rhythm or dyslexia in the original GWAS reports[17,18] and three additional education-related traits[28] (Supplementary Table 13). To reduce the statistical burden of multiple testing correction in our consequent analyses, we subset this initial set of 88 traits on the basis of their levels of genetic correlation among themselves. To do so, we estimated pairwise genetic correlations, and identified the most highly correlated traits ($|r_g| > 0.80$; Supplementary Fig. 5). We then performed hierarchical clustering, obtaining one representative trait from each cluster of highly correlated traits (Supplementary Fig. 6). This approach yielded 49 traits that were relatively genetically independent (see Methods for details), for which we estimated the genetic correlations with $F_{gRI-D}$, and with the summary statistics obtained by the IPM (Supplementary Fig. 7 and Supplementary Table 14). Genetic correlations between $F_{gRI-D}$ and the assessed traits ranged from −0.56 to 0.46, and mostly lay between the genetic correlation estimates for independent factors (Supplementary Fig. 7), supporting that $F_{gRI-D}$ indeed captures the common genetic factor of rhythm impairment and dyslexia. We found significant negative correlations between $F_{gRI-D}$ and non-word repetition ($r_g$ (s.e.m.) = −0.513 (0.099), $P_{FDR} = 7.03 \times 10^{-7}$) and phoneme awareness ($r_g$ (s.e.m.) = −0.562 (0.058), $P_{FDR} = 3.78 \times 10^{-21}$), validating the $F_{gRI-D}$ construct's link to reading- and language-related traits. Positive genetic correlations were observed for attention deficit hyperactivity disorder ($r_g$ (s.e.m.) = 0.237 (0.029), $P_{FDR} = 3.69 \times 10^{-15}$), autism spectrum disorder ($r_g$ (s.e.m.) = 0.075 (0.035), $P_{FDR} = 4.529 \times 10^{-2}$) and insomnia ($r_g$ (s.e.m.) = 0.200 (0.027), $P_{FDR} = 6.04 \times 10^{-13}$), suggesting shared genetic liability with neuropsychiatric traits that have been phenotypically linked to rhythm[47]. In total, $F_{gRI-D}$ showed significant ($P_{FDR} < 0.05$) genetic correlations with 37 of the 49 selected psychiatric/neurological/behavioural traits with varying magnitudes and directions, including attention deficit hyperactivity disorder, Parkinson's disease, health satisfaction and loneliness ($|r_g|$ median of 0.146 and

**Fig. 2 | Manhattan plots for univariate and mvGWASs and heterogeneity, including examples of highly homogeneous and heterogeneous loci in $F_{gRI-D}$ results. a**, Manhattan plots of dyslexia, musical rhythm GWASs, the common factor ($F_{gRI-D}$) mvGWAS and the heterogeneity between dyslexia and musical rhythm impairment GWASs ($Q_b$). The *y* axes show −$\log_{10}(P)$ values; the *x* axes show chromosomal positions and datapoints represent SNPs. Dyslexia and musical rhythm GWASs were previously performed and published and are included here for illustration purposes. The mvGWAS results and heterogeneity estimates were obtained using a CPM in genomic SEM. Loci that pass the genome-wide significance threshold ($P < 5 \times 10^{-8}$) in $F_{gRI-D}$ and $Q_b$ Manhattan plots are listed in Supplementary Tables 2 and 3. The red lines correspond to the genome-wide significance threshold ($P < 5 \times 10^{-8}$). $N_{rhythm impairment} = 606,825$, $N_{dyslexia} = 1,138,870$.

**b**, LocusZoom plots of example homogeneous and heterogeneous loci, chosen on the basis of $Q_b$ *P* values. The *y* axes show −$\log_{10}(P)$ values, and the *x* axes show chromosomal positions of each SNP. Each triangle represents a SNP and the direction of the triangle indicates the sign of the GWAS effect (upwards, positive effects; downwards, negative effects). Colour codes correspond to the linkage disequilibrium with the lead SNP. The SNP loadings on D and RI indicate GWAS effect sizes and directions (left), whereas the SNP loadings in the SEM diagrams show IPM effect sizes and directions (right) of two example SNPs, reflecting the homogeneous versus heterogeneous architecture of the example loci, respectively. $F_g$, common genetic factor of dyslexia and musical rhythm impairment; D, dyslexia GWAS; RI, musical rhythm impairment GWAS; cM, centimorgan; u variables, residual variance not explained by the common factor.

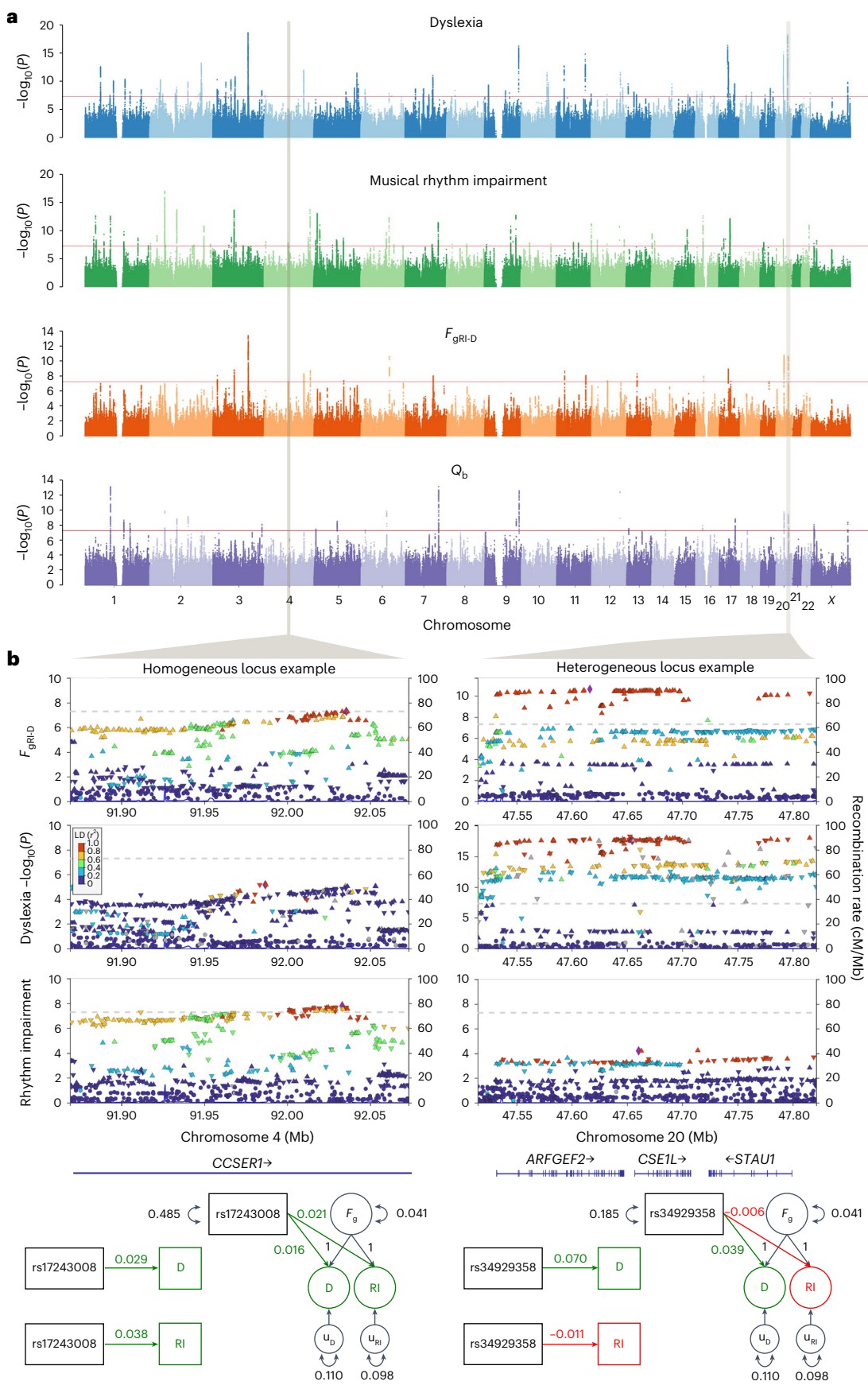

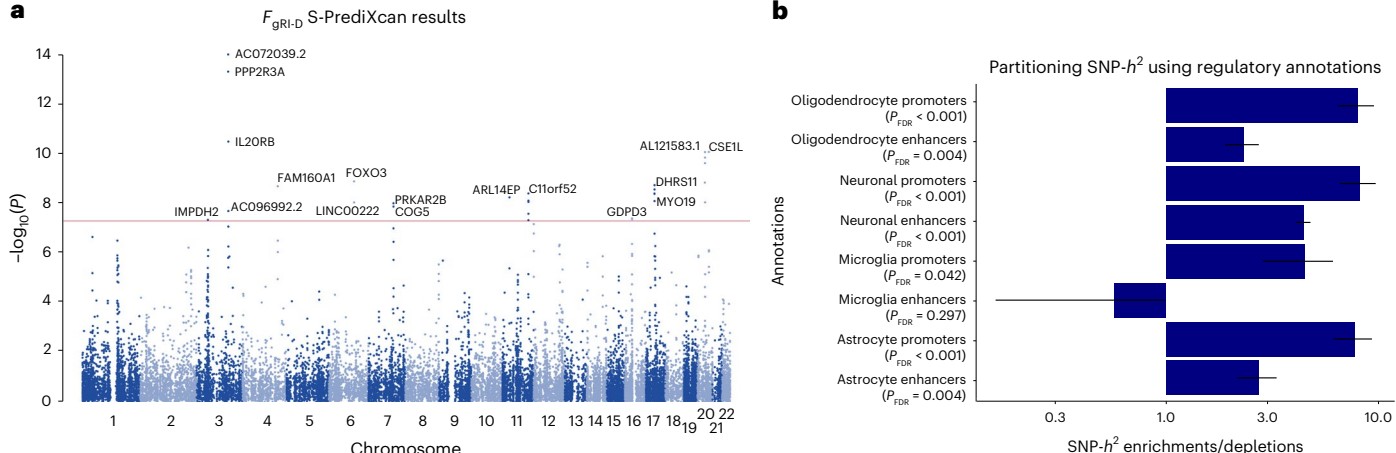

**Fig. 3 | S-PrediXcan and LDSC partitioned heritability results for regulatory brain cell type annotations. a**, Manhattan plot showing TWAS results on 13 brain tissue and whole-blood tissues. Each data point corresponds to a gene–tissue pair. The $y$ axis shows $-\log_{10}(P)$ values, and the $x$ axis shows chromosomal positions of each gene–tissue pair. The most significant gene–tissue association pair is shown for each gene. The red line corresponds to the genome-wide significance threshold ($P < 5 \times 10^{-8}$). **b**, Bar plots showing LDSC SNP-$h^2$ enrichment and depletion estimates for the common genetic factor of musical rhythm impairment and dyslexia ($F_{gRI-D}$; $N_{rhythm impairment} = 606,825$ and $N_{dyslexia} = 1,138,870$) on the $x$ axis and eight brain cell type-specific regulatory annotations on the $y$ axis. $P$ values were estimated using a two-sided test and FDR corrected for eight tests. The exact enrichment or depletion $P$ values are reported in Supplementary Table 12. Error bars represent s.e.m., with whiskers extending one standard error below or above the SNP-$h^2$ enrichment or depletion estimate.

range of 0.06–0.56). Consistent with the ARRH, the directionality of genetic correlations suggest that decreased rhythm impairment/dyslexia risk may be associated with resilience to certain neuropsychiatric disorders. These genetic correlations also reflect a shared genomic architecture underlying rhythm, dyslexia and social traits, showing that social function and co-evolution hypotheses of rhythm and communication skills[48–50] are plausible from a genetic perspective. Future work will be needed to disentangle possibly shared genomic substrates of the evolution of social interaction, language and music.

Even though reading is a recent human innovation, it recruits language-related brain circuits[51,52], which have undergone biological evolution on the lineage leading to humans. Similarly, dyslexia manifests overtly as a reading or spelling disorder, yet in many cases this reflects underlying deficits in aspects of oral language (for example, phonological awareness)[19–22]. Given this link between spoken language and reading, and in light of theoretical frameworks positing co-evolution of rhythm and language-related skills in humans[10,48–50,53], we leveraged genomic methods to investigate the evolution of the overlap between rhythm and the reading-related aspect of language over a range of timescales (Fig. 4a). We first performed LDSC partitioned heritability analysis using five evolutionary annotations tagging foetal brain human-gained enhancers[54], Neanderthal introgressed alleles[55], archaic deserts[56] and primate-conserved and accelerated regions[57] (Fig. 4a). This revealed significant SNP heritability depletions in Neanderthal introgressed alleles, and significant enrichments in primate-conserved regions for all traits (Fig. 4b and Supplementary Table 15), in line with findings for many other complex human traits[58,59]. We then used the SBayesS function of the GCTB package[60] to probe the effect size-minor allele frequency relationship ($\hat{S}$)—an essential component of the complex trait genetic architecture influenced by natural selection[60]. Similar to most cognitive and behavioural traits[60], we found moderate levels of negative selection acting on $F_{gRI-D}$ ($\hat{S}$ (s.d.) = −0.51 (0.05)) and the independent factors of dyslexia $\hat{S}$ (s.d.) = −0.47 (0.06)) and rhythm impairment ($\hat{S}$ (s.d.) = −0.49 (0.06)) (Fig. 4d and Supplementary Table 16). Next, we performed MAGMA gene set analysis[61] to investigate links between genes that are co-located with various evolutionary annotations, spanning a timescale from ~8 million to ~35,000 years ago, which were not testable via partitioned heritability analysis owing to low SNP coverage (Methods). Specifically, we tested whether genetic variation

associated with $F_{gRI-D}$ was enriched in genes that overlap with four evolutionary annotations (Supplementary Tables 17–20): (1) ancient selective sweep sites[62], (2) human accelerated regions[63–66], (3) differentially methylated regions (DMRs) between anatomically modern humans and archaic humans[67] and (4) DMRs between anatomically modern humans and chimpanzees[67]. These gene set-based analyses did not yield any significant enrichment signals (Supplementary Table 21), indicating an absence of evidence for associations between $F_{gRI-D}$ and these four annotations. We then extended our MAGMA gene set enrichment analysis to look for potential links between $F_{gRI-D}$ and genomic substrates of songbird vocal learning, in line with theoretical predictions[10], and prior associations with the genetic architecture of beat synchronization[68]. To do so, we used nine gene sets that were curated by Gordon et al.[68] and therein converted to human homologues for the purposes of gene enrichment analyses; each set represents differential gene expression patterns associated with vocal learning phenotypes (for example, song versus silence or number of motifs sung) in Area X and other key regions of zebra finch neural circuitry related to song learning. Intriguingly, we found significant enrichments in four Area X-related gene sets (Supplementary Table 22) using the $F_{gRI-D}$ summary statistics. These findings may suggest overlapping molecular mechanisms between songbird vocal learning, human rhythm and human language, supporting theories of cross-species convergent evolution of vocal learning and beat perception[10,69].

To follow up the significant partitioned SNP heritability enrichments in primate-conserved regions, we investigated the association between $F_{gRI-D}$ mvGWAS $P$ values and per-SNP primate phastCons scores[57] for 38,164 clumped SNPs ($P < 0.05$, $r^2 < 0.06$) from $F_{gRI-D}$ summary statistics (Fig. 4c), and found that one of the $F_{gRI-D}$ genome-wide significant ($P < 5 \times 10^{-8}$) hits, rs10891314, had an exceptionally high phastCons score, probably because it is a missense variant (Fig. 4c). We zeroed-in on this genome-wide significant hit as an example locus and dissected patterns of $Q_b$, and conservation or accelerated evolution in primates (Fig. 4e), confirming the sharp increase in conservation rate for the SNP rs10891314. The Human Genome Dating Atlas[70] estimates this polymorphism to be 11,199 generations old (95% CI), corresponding to ~280,000 years ago assuming 25 years per generation, around the time period when the oldest known *Homo sapiens* fossils have been dated[71]. rs10891314 is located in the *DLAT* gene, which is associated with

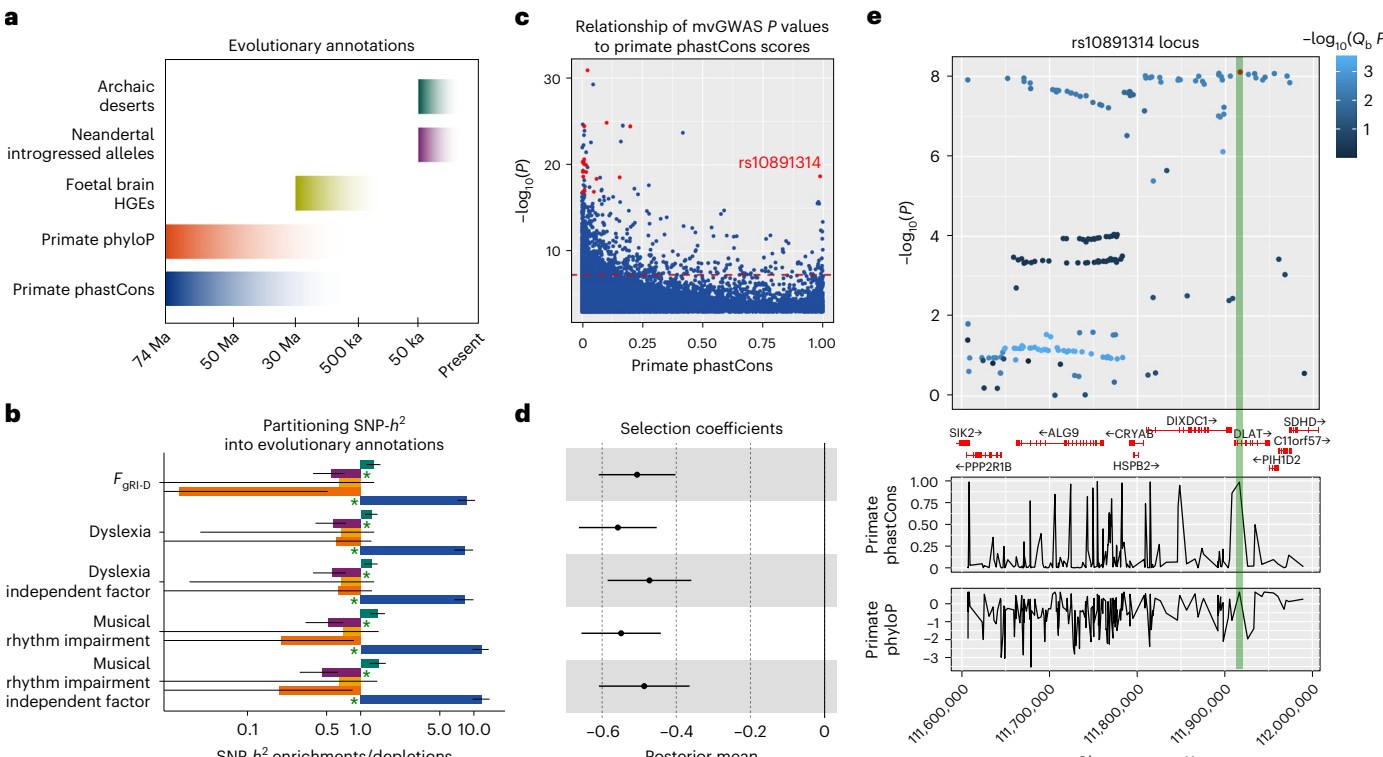

**Fig. 4 | Evolutionary analyses of dyslexia, rhythm impairment, $F_{gRI-D}$ and independent factors. a**, The timescales covered by the evolutionary annotations that we used. HGEs, human-gained enhancers; Ma, million years ago; ka, thousand years ago. **b**, LDSC partitioned SNP-$h^2$ enrichment and depletion estimates on the $x$ axis and annotation-trait pairs on the $y$ axis ($N_{rhythm\,impairment}$ = 606,825 and $N_{dyslexia}$ = 1,138,870). Colour coding of the bars corresponds to evolutionary annotations in **a**. $P$ values were estimated using a two-sided test. Green asterisks indicate statistical significance after FDR correction for 25 tests ($P_{FDR}$ < 0.05). Error bars represent s.e.m., with whiskers extending one standard error below or above the SNP-$h^2$ enrichment or depletion estimate. The exact enrichment or depletion $P$ values are reported in Supplementary Table 15. **c**, A scatter plot showing the association between $F_{gRI-D}$ mvGWAS $-\log_{10}(P)$ values on the $y$ axis and primate phastCons scores on the $x$ axis. Lead SNPs in 17 genome-wide significant ($P$ < 5 × 10⁻⁸) loci are highlighted

in red (one genome-wide significant lead SNP does not have a phastCons score and is not shown here). The dashed red line indicates the mvGWAS genome-wide significance threshold ($P$ < 5 × 10⁻⁸). **d**, The GCTB SBayesS selection coefficient estimates as posterior means shown on the $x$ axis, for each trait on the $y$ axis (note that the $y$ axis is shared with **b**). The error bars represent s.d., with whiskers extending one s.d. below or above each coefficient. The vertical line is fixed at zero. ($N_{rhythm\,impairment}$ = 606,825 and $N_{dyslexia}$ = 1,138,870). **e**, The results of a manual look-up of the SNP rs10891314, showing its co-localization with *DLAT*. Each data point is a SNP, with the $y$ axis showing $F_{gRI-D}$ mvGWAS $-\log_{10}(P)$ values and the $x$ axis showing genomic locations. Colour coding reflects $Q_b$ (heterogeneity) scores. The primate phastCons and primate phyloP graphs show patterns of conservation and accelerated evolution within the locus. rs10891314 is shown in red, and its phastCons and phyloP scores are shown with the green bar.

a rare neurodevelopmental disorder, pyruvate dehydrogenase E2 deficiency. This condition is characterized by neurological dysfunction, dystonia and learning disability mainly appearing during childhood[72]. *DLAT* is highly conserved and loss-of-function intolerant (probability of loss-of-function intolerance 6.68) (ref. [73]), which makes this particular missense variant an interesting candidate for increasing susceptibility to rhythm impairment and dyslexia.

After assessing evolutionary signatures on $F_{gRI-D}$ at the genome-wide and SNP levels, we extended our investigations of rhythm–language co-evolution by integrating with independent data from neuroimaging genetics. To do so, we estimated local genetic correlations between $F_{gRI-D}$ and fractional anisotropy (FA) measures of five left hemispheric white matter tracts (Supplementary Fig. 8 and Supplementary Table 23), involved in the dorsal or ventral streams of spoken language, and theorized as key components of rhythm–language convergent evolution[10,69]. Using local analysis of (co)variant association (LAVA)[74], we identified a significant local genetic correlation between $F_{gRI-D}$ and the left hemispheric superior longitudinal fasciculus (SLF) I ($r_g$ = 1, $P_{FDR}$ = 0.02) (Supplementary Table 24) on a ~2 Mb region on chromosome 20 (30,569,660–32,484,506), which encompasses several genes including *EFCAB8*, *BAK1P1* and *SUN5* (Supplementary Fig. 9). SLF-I is the dorsal division of SLF connecting the

superior parietal and superior frontal lobes[75] (Supplementary Fig. 8). SLF has been shown to have functional links to musical rhythm[76,77], and SLF-I subdivision is involved in the regulation of motor behaviour[78,79]. This finding is partially consistent with the hypothesized role of the dorsal stream in supporting co-evolution of phonological processing and beat synchronization[4].

## Discussion

We showed robust genetic correlations among musical rhythm, dyslexia and a number of reading- and language-related traits, providing genetic evidence for the ARRH. Traits, such as non-word repetition, phoneme awareness and having better language skills than mathematics at school, showed significant positive correlations with rhythm skills, which makes these particular traits interesting candidates to study in future genetic studies of shared biology of rhythm and language. Importantly, we also found a significant correlation between rhythm and language resting-state functional connectivity, suggesting shared genetic and neuronal architecture for rhythm and reading-related aspects of language.

The bivariate genomic SEM approach that we developed allowed us to identify genetic overlaps between rhythm impairment and dyslexia and to present a map of homogeneous and heterogeneous genetic

effects, shedding light on patterns of pleiotropy between the two[1]. Among 18 genome-wide significant loci associated with the common factor of dyslexia and rhythm impairment, the strongest mvGWAS signal comes from a locus tagged by the SNP rs28576629 that is mapped to *PPP2R3A*, a gene implicated in the negative control of cell growth and division, suggesting a putative role for this gene in dyslexia and rhythm impairment prevalence[80]. Given that we validated our common factor results using two additional mvGWAS methods, we believe that the shared genetic factor that we captured represents a solid first glimpse into the shared genetics of dyslexia and rhythm impairment. Results of this kind might potentially contribute, together with information on other risk factors, towards improved diagnostics of individuals' propensity for reading- and rhythm-related problems, to enable special educational support. However, given the highly polygenic and environment-dependent nature of behavioural traits, the early risk identification power of our results remains to be explored[81].

Our post-mvGWAS analyses enhance understanding of the aetiology of rhythm and language (on which reading depends) by revealing intricate links across rhythm impairment, dyslexia and various aspects of evolutionary past and neurobiological function, including gene expression in brain tissue, brain cell type-specific gene regulation and a local genetic correlation with a tract linked to regulation of higher aspects of motor behaviour[75]. Our TWAS results validated the association between *PPP2R3A* and the common factor of dyslexia and rhythm impairment, and narrowed down the overall relevance of this gene for rhythm and reading skills into its expression profile in cerebellum and putamen. We believe that our TWAS results constitute a potentially important gene–tissue pair list to study the links between genetic variants, gene expression in the brain and subsequent effects on neurodevelopment using experimental assays. Our SNP heritability enrichment results in cell type-specific regulatory regions point to multiple brain cell types for follow-up work, without pinning down a specific neuronal or non-neuronal brain cell type, indicating a complex cellular composition in the brain supporting rhythm and language. Heritability enrichment signals in brain-specific regulatory regions provide additional evidence that the dyslexia and rhythm GWASs largely capture the neurodevelopmental aspects of these traits. The genome-wide genetic correlation analysis between the common factor and a set of behavioural traits yielded two important findings. First, the genetic correlation directions and magnitudes point to a link between rhythm impairment and dyslexia risks, and neuropsychiatric disorders, in line with the ARRH. Second, language-related traits, such as non-word repetition and phoneme awareness, yielded the strongest genetic correlations with the common factor of dyslexia and rhythm impairment, further supporting the statistical validity and language or reading relevance of $F_{gRI-D}$.

The evolutionary analyses aimed to provide empirical genetic data as groundwork towards understanding potential evolutionary forces acting jointly on human rhythm and language-related skills[53,82]. Our significant SNP heritability depletion findings in Neanderthal introgressed alleles is in line with findings for other complex human traits[58], indicating a reduced contribution of Neanderthal alleles to reading- and rhythm-related skills. Similarly, heritability enrichment findings in primate-conserved regions converge with studies that previously found significant enrichments in these loci for complex disease traits[59]. We also noted a trend for the majority of mvGWAS lead SNPs to have lower primate conservation scores, indicating weaker constraint on these variants in the primate clade, which may have relevance for the evolution of language and rhythm-related traits on the human lineage. This observation lacks statistical confirmation, but would be in line with prior behavioural and neural findings showing a lack of complex musical rhythm detection and synchronization in species, such as macaque monkeys[83]. Interestingly, one mvGWAS hit locus, mapped to the *DLAT* gene, stood out as strongly conserved among primates, which makes this gene a potential candidate for

future experimental investigations in this area. Our analyses showed significant enrichments of $F_{gRI-D}$-associated variants in gene sets curated from transcriptome studies of songbird vocal learning, specifically in a key nucleus of the zebra finch brain, Area X. This link between human genetic variants shaping human reading and rhythm skills, and genes involved in songbird song production (for example, number of motifs sung) in Area X is in line with prior literature[68], further supporting the importance of shared genetic substrates. Finally, the significant local genetic correlation that we identified between SLF-I and $F_{gRI-D}$-associated variants in a ~2 Mb region on chromosome 20 represents an interesting example of potential pleiotropic associations between language- and musical rhythm-related traits and white matter microstructure. This finding is particularly interesting as SLF-I is involved in motor behaviour regulation[78,79], suggesting the presence of shared genetic and neuroanatomical elements between motor aspects of language and musical rhythm. It is also plausible that the shared genetic factors underlying language- and musical rhythm-related skills influence a broader range of cognitive processes rather than being confined to the intersection between language and musical rhythm. Here, we note that future local and genome-wide genetic correlation analyses between $F_{gRI-D}$ and a larger selection of neuroanatomical traits (for example, anterior and posterior segments of the arcuate fasciculus, inferior fronto-occipital fasciculus) and other imaging modalities are necessary to reveal the shared genetics and neuroanatomy of language- and musical rhythm-related traits.

Our study has several limitations. First, the original dyslexia and rhythm GWASs were performed on European ancestry samples owing to data availability reasons. The lack of large-scale GWASs in non-European populations and the strong sampling bias in large cohorts towards European ancestry individuals hinder a global picture of the genetic architecture of these traits, weaken the replicability of GWAS findings in diverse populations[84] and limit the interpretability of post-GWAS evolutionary analysis results. This is especially true for behavioural and psychiatric traits that are more prone to be affected by cultural, socioeconomic and environmental factors, which is also reflected in the weaker transferability of polygenic scores across ancestries for such traits[85]. We believe that important steps to solve this will include encouraging and contributing to the generation of large-scale databases with genotype or phenotype data of individuals from diverse genetic ancestries, disentangling the unique gene–environment interactions in other populations, developing GWAS methods and study designs to more carefully take potential confounders into account and improving the reliability of behavioural phenotype measurement techniques. Second, the self-report-based phenotype descriptions in the original GWASs are not ideal measures, but rather represent robust validated proxies that uniquely enable scaling up of data collection to very large cohorts. There is a compromise between the practical convenience of self-report-based data collection from hundreds of thousands of individuals, which is extremely challenging using task-based measurements, and introducing self assessment-related uncertainties to the data. We note that the phenotype in the dyslexia GWAS was not simply self assessment but rather self-report of having received a positive dyslexia diagnosis, and that the genetic architecture was found to be stable across the lifespan, with a genetic correlation of 0.97 between younger (<55 years) and older subgroups (>55 years) of participants[18]. Moreover, the construct validity of the rhythm GWAS phenotype is supported by associations between the self-reported and directly measured rhythm skills[17], and polygenic scores trained on the rhythm summary statistics correlate with scores on a rhythm discrimination test[86,87], further supporting the view that genetic signals associated with self-reported beat synchronization ability are an appropriate reflection of rhythm-related skills. Moreover, the genetic correlations between dyslexia, rhythm impairment, $F_{gRI-D}$ and other speech and language-related phenotypes suggest that the original GWASs largely capture relevant genetic factors. Third, even though

potential confounders, such as age and sex, were included as covariates in the dyslexia and rhythm GWAS regression models, we cannot fully exclude residual effects of such factors and other confounders. The original dyslexia GWAS addressed the impacts of age (as noted above) and sex by performing sensitivity analyses using age- and sex-specific GWAS, which showed little effect of either of these two factors on GWAS results[18]. Fourth, the majority of genetic factors shaping human language and musical rhythm skills are probably fixed in all human populations. Hence, post-GWAS evolutionary analyses, which leverage present-day variation to probe links between genetic variants and evolutionary annotations, are not ideal to study the origins of traits that probably emerged at earlier periods of hominin evolution. Methodological developments in the complex trait evolution field, and the integration of ancient DNA data from older timepoints of human evolution into polygenic selection analysis methods would greatly help to resolve these challenges in future studies. Finally, relationships between heritability and evolution are quite complex. The individual contributions of common genetic variants to heritability are jointly shaped by selection and demography[88], which limits the evolutionary interpretation of heritability-based methods.

Despite such constraints, our study represents a step towards characterizing the shared genetic architecture between rhythm- and language-related traits, and provides a valuable analytic pipeline tackling the shared genetics and evolution of rhythm and reading-related aspects of language from various angles. We reveal complex links across common DNA variants, genes, genomic loci, white matter structures and human behaviour, making a first set of links across the immensely long causal chain spanning these layers. Developing and applying more sophisticated methods to dissociate environmental confounds from genetics will allow future studies to obtain a better understanding of the genetics and evolution of human language and musicality.

## Methods

### GWAS summary statistics
Beat synchronization and dyslexia GWAS summary statistics[17,18] were obtained from 23andMe Inc., a customer genetics company. Both GWASs were performed on European ancestry individuals through online participation. All participants provided informed consent according to 23andMe's human subject protocol, which is reviewed and approved by the external Association for the Accreditation of Human Research Protection Programs, Inc.-accredited institutional review board, Ethical and Independent Review Services, a private institutional review board (http://www.eandireview.com). The 23AndMe sample prevalence of dyslexia is 4.6% ($N_{total}$ = 1,138,870, mean age 51 years), and sample prevalence of beat synchronization is 92% ($N_{total}$ = 606,825, mean age 52 years). Summary statistics files were reformatted and harmonized to include required columns (for example, SNP ID, beta, beta s.e.m. and $P$ value) for each mvGWAS tool following the guidelines in the original publications of each tool. To obtain rhythm impairment summary statistics, effect sizes in the binary beat synchronization GWAS summary statistics were multiplied by −1 so that the effect directions were reversed. This yielded a set of GWAS summary statistics comprising SNP effects contributing to rhythm impairment, which was used for the subsequent mvGWAS analysis with dyslexia. We applied GC correction to both sets of summary statistics for all non-LDSC-based analyses. For LDSC-based analyses (including genomic SEM), uncorrected summary statistics were used as input, as GC correction biases the LDSC SNP heritability estimates downwards. The resulting set of summary statistics from genomic SEM was GC corrected.

### SNP heritability and genetic correlation estimations
We used LDSC[25] (v1.0.1) to estimate the SNP heritabilities and genetic correlations. For rhythm impairment and dyslexia, we estimated the total SNP heritability on a liability scale using population and sample prevalence information from the original studies (sample prevalence of 0.045

for dyslexia and 0.085 for rhythm impairment, and a population prevalence of 0.050 for dyslexia and 0.048 for rhythm impairment). Genetic correlations were estimated using bivariate LDSC between rhythm, dyslexia, GenLang quantitative reading or language-related traits[27], Danish School Grades GWAS[28] and all external summary statistics except for the planum temporale asymmetry and the language resting-state functional connectivity, which were assessed as described below.

To estimate genetic correlations between rhythm and planum temporale asymmetry[30] and between rhythm and language resting-state functional connectivity[29], we used an approach proposed by Naqvi et al.[89] applicable to unsigned multivariate statistics, as the mvGWAS effect sizes or beta values, which are required to run genetic correlation analysis using LDSC, were not available for these traits. We evaluated the amount of shared signal between each pair of GWASs by estimating the Spearman correlation of the average SNP $P$ values within approximately independent linkage disequilibrium (LD) blocks[90]. We first filtered the genome-wide SNPs using the HapMap3 reference panel without the major histocompatibility complex region (https://github.com/bulik/ldsc). We then split the genome-wide SNPs into 1,703 approximately independent blocks[90]. For each approximately independent LD block, we computed the average SNP $-\log_{10}(P)$ value. We then estimated a rank-based Spearman correlation using the averaged association value ($n$ = 1,703) for each LD block. A standard error of the Spearman correlation was estimated using statistical re-sampling with 10,000 bootstrap cycles with replacement from the 1,703 LD blocks.

### mvGWAS
To investigate the shared genetic variance of rhythm impairment and dyslexia, we performed mvGWASs using three tools: genomic SEM[26], $N$-weighted GWAMA[33] and CPASSOC[34]. These tools use GWAS summary-level data and account for genetic correlation and sample overlap using the cross-trait LD score regression intercept.

### Genomic SEM (common and independent pathway models).
First, we reformatted our summary statistics for LDSC (munged) and genomic SEM, following standard guidelines (https://github.com/GenomicSEM/GenomicSEM/wiki). We then used the multivariable extension of LDSC to estimate the 2 × 2 empirical genetic covariance matrix between rhythm impairment and dyslexia and their associated sampling covariance matrix. We specified a measurement model (Supplementary Fig. 1), where a shared genetic factor ($F_{gRI-D}$) was estimated to capture the observed genetic covariance between rhythm impairment and dyslexia. Given that the number of observed parameters for any 2 × 2 covariance matrix equals 3, we constrained all paths between $F_{gRI-D}$ to 1. The final CPM was fit to a genetic covariance matrix that incorporates the tested SNP (Supplementary Fig. 1). SNPs were regressed from $F_{gRI-D}$, and residuals were freely estimated. The 1,000 Genomes Phase 3 reference panel[91] was used as the reference panel to calculate SNP variance across traits. Effective population size per-GWAS was calculated as $4 \times N_{cases} \times (1 - N_{cases}/N_{total})$. Both the reference panel and effective population sizes were then fed into the sumstats function and summary statistics were prepared for the meta-analysis. We applied genomic correction to the CPM results on the basis of the genomic inflation index estimated by LDSC ($\lambda_{GC}$ = 1.62; Supplementary Fig. 2). The final IPM, was fit to the same matrices incorporating the SNP effects, but with the SNP effect being directly regressed from the traits. The final bivariate heterogeneity score, $Q_b$, was obtained by subtracting by a chi-squared difference test, where the chi-squared of the IPM is subtracted from the chi-squared of the CPM ($Q_b = \chi^2_{CPM} - \chi^2_{IPM}$) (ref. 31). A high $Q_b$ value indicates that the association between the SNP and rhythm impairment or dyslexia is not well accounted for by the factor $F_g$. We then used the intersect function of bedtools (v. 2.29.2)[92] to identify the overlaps between genome-wide significant ($P < 5 \times 10^{-8}$) $Q_b$ (Supplementary Table 3) and CPM loci (Supplementary Table 2), as well as ±1 Mb surroundings of each CPM locus.

**CPASSOC.** Following the CPASSOC manual[34], we used the median sample size for each summary statistics file, as 23andMe SNPs can have varying sample sizes. We removed SNPs with a $Z$-score larger than 1.96 or less than −1.96, and extracted a $2 \times 2$ genetic correlation matrix for dyslexia and rhythm impairment. Then we generated an $M \times K$ matrix of summary statistics where each row represented a SNP, and two columns represented dyslexia and rhythm impairment $Z$-scores. We finally performed the $S_{hom}$ test, and obtained a vector of $P$ values for $M$ SNPs using the pchisq function in R (4.0.3).

**GWAMA ($N$-weighted).** To account for sample overlap, we first generated a matrix of cross-trait intercepts using the intercepts of LDSC genetic correlations between dyslexia and rhythm impairment summary statistics. We then performed $N$-weighted GWAMA by feeding the cross-trait intercept matrix and a vector of SNP heritabilities of each trait using the multivariate_GWAMA function.

**TWAS**
We conducted a TWAS using S-PrediXcan framework[37] and the joint-tissue imputation (JTI) TWAS derived models from GTEx v8 tissues[35]. PrediXcan predicts gene expression from the genotype profile of each individual by using the JTI model weights, which were trained on GTEx[93], and validated on PsychEncode[94] and GEUVADIS[95]. These SNP expression weights represent the correlations between SNPs and gene expression levels. To overcome the requirement for individual-level genotype data, Barbeira et al.[37] derived a mathematical expression, implemented in S-PrediXcan framework, which effectively yields similar outcomes to PrediXcan using GWAS summary statistics. S-PrediXcan and JTI weights account for LD and collinearity problems owing to high expression correlation across tissues[35]. We filtered the 17q21.31 inversion region (~1.5 Mb long), which has multiple phenotypic associations with brain-related traits[96] to minimize the impact of this high-LD region on our results. We then corrected TWAS $P$ values for 192,905 gene–tissue pairs, and used $Z$-scores and $P_{FDR}$ of the significant ($P_{FDR} < 0.05$) pairs to assess gene–$F_{gRI-D}$ associations.

**Gene set enrichment and pathway analyses**
We used PANTHER to run statistical overrepresentation analysis in three GO and three PANTHER GO-Slim terms (biological process, molecular function and cellular component)[38–41] with 315 unique genes that we obtained from TWAS. We used 20,102 genes that we tested in TWAS as the background gene set. Results were FDR corrected for all GO and GO-Slim terms ($n = 15,028$).

**LDSC partitioned heritability with cell type-specific annotations**
We used eight human genome annotations by Nott et al.[45] tagging promoter and enhancer regions of neurons, oligodendrocytes, microglia, and astrocytes using LDSC partitioned heritability analysis[44] following the guidelines in the LDSC Wiki page (https://github.com/bulik/ldsc/wiki/Partitioned-Heritability). All enrichment analyses were controlled for the baselineLD model v2.2. Enrichment $P$ value results were FDR corrected for eight tests.

**Genetic correlations using GWAS summary statistics from neuropsychiatric or behavioural phenotypes**
We first compiled 88 traits that were significantly genetically correlated either with rhythm (impairment) or dyslexia in the original respective GWAS papers[17,18]. We filtered these traits to avoid unnecessary multiple testing burden and to focus on genetically independent phenotypes. We identified 46 traits that are more than ±80% genetically correlated with at least one other trait. Next, we created a distance matrix from the correlation estimates and performed hierarchical clustering using Ward's method[97] as the linkage method, which maximizes the within-cluster homogeneity to identify trait clusters. We identified seven clusters using the so-called elbow method, and chose the most informative and representative trait for each cluster on the basis of the highest correlation between traits and the cluster principal component. We added these seven cluster-representative traits to the remaining 42 traits and used LDSC to estimate genetic correlations with $F_{gRI-D}$ and two independent factors. Genetic correlation $P$ values were FDR corrected for 49 tests.

**Partitioned heritability analysis with custom evolutionary annotations**
We used LDSC[25] (v1.0.1) to estimate partitioned SNP heritability enrichments or depletions in foetal brain human-gained enhancers[54], Neanderthal introgressed alleles[55], archaic deserts[56], conserved loci in the primate phylogeny (Conserved_Primate_phastCons46way annotation from baselineLD) and genomic loci that have a primate phyloP score[57] less than −2 (presumably suggesting accelerated evolution). All annotations were controlled for baselineLD model v2.2. Foetal brain human-gained enhancers were also controlled for foetal brain active regulatory elements from the Roadmap Epigenomics Consortium database[98].

**MAGMA gene set analysis with custom evolutionary and songbird vocal learning gene lists**
We compiled four additional evolutionary genomic annotations for MAGMA gene set analysis[61], which cover timescales from ~8 million years ago to ~35,000 years ago: ancient selective sweeps[62], human accelerated regions[63–66], anatomically modern human-derived DMRs[67] and human versus chimpanzee DMRs[67]. These annotations either tag regulatory or selective sweep sites, and cover less than 1% of the total number of well-imputed SNPs in 1,000 Genomes Phase 3 reference panel, which makes them unsuitable for LDSC partitioned heritability analysis. We listed the genes that fall within ±1 kilobase of each locus tagged by each annotation, and filtered these initial gene lists for protein-coding genes using the National Center for Biotechnology Information's (NCBI) hg19 genome annotation[99]. The resulting protein-coding gene lists were used for MAGMA gene set enrichment analysis for $F_{gRI-D}$ summary statistics. We first performed gene annotation by integrating SNP locations from the summary statistics, and gene locations from NCBI hg19 genome annotations. We then performed a gene analysis using SNP $P$ values and 1,000 Genomes Phase 3 European panel[91]. We finally applied a gene set analysis using results from gene annotation and gene analysis, and four gene-sets. Enrichment $P$ values were FDR corrected for four tests. In the second part of our MAGMA analysis, we used nine additional songbird brain-expressed gene sets from Gordon et al.[68] and performed a separate gene set enrichment analysis for $F_{gRI-D}$. Enrichment $P$ values were FDR corrected for nine tests.

**Genome-wide negative selection estimation**
We performed SBayesS analysis on the rhythm impairment, dyslexia, $F_{gRI-D}$, and two independent factor GWAS summary statistics using the GCTB software (version 2.02)[60] to quantify the level of negative selection acting on these traits. SBayesS estimates total SNP heritability, polygenicity and the relationship between variants' minor allele frequencies and effect sizes, and generates a genome-wide negative selection metric ($S$), which ranges from 0 to −1. $S$ estimates that are closer to −1 are interpreted as a sign of strong negative selection[50], whereas estimates closer to 0 can suggest positive selection[60].

**LAVA local genetic correlations with white matter connectivity measures**
To identify local regions of the genome that might make shared contributions to rhythm, language and evolutionarily relevant brain circuitry, we tested local genetic correlations between $F_{gRI-D}$ and white matter connectivity measures. We performed GWASs of selected brain imaging

traits using data from the UK Biobank[100]. For these GWASs, UK Biobank data first underwent sample and genetic quality control and brain imaging data processing, followed by genome-wide association analysis.

**Sample quality control.** This study used the UK Biobank February 2020 release (research application number: 79683). All participants provided informed consent, and the study was approved by the North West Multi-Centre Research Ethics Committee. For individuals with both diffusion-weighted magnetic resonance imaging (MRI) and genotyping data, we excluded participants with unusual heterozygosity (principal components corrected heterozygosity >0.19), high missingness (missing rate >0.05), sex mismatches between genetically inferred sex and self-reported sex as reported by Bycroft et al.[100]. We further restricted our analyses to individuals with white British ancestry as defined by Bycroft et al.[100] to avoid any possible confounding effects related to ancestry. This resulted in 31,465 individuals (mean age of 55.21 years old, range 40–70 years old, 16,497 females) passing the sample quality control.

**Genetic quality control.** The imputed genotypes were obtained from the UK Biobank portal. These data underwent a stringent quality control protocol. We excluded SNPs with minor allele frequencies below 1%, Hardy–Weinberg $P$ value below $1 \times 10^{-7}$ or imputation quality INFO scores below 0.8. Multi-allelic variants that cannot be handled by many programs used in genetic-related analyses were removed. This resulted in 9,422,496 autosomal SNPs that were analysed in the GWAS.

**Neuroimaging phenotypes.** The diffusion-weighted MRI data were acquired from a 3-T Siemens Skyra scanner using the following parameters: isotropic voxel size (resolution) of $2 \times 2 \times 2$ mm, five non-diffusion-weighted images ($b = 0$ s mm$^{-2}$), diffusion-weighting of $b = 1,000$ and $2,000$ s mm$^{-2}$ with 50 directions each, and acquisition time of 7 min. Whole-brain diffusion-weighted MRI scans were acquired in vivo, and fed into diffusion tensor imaging (DTI) fitting toolbox to assess brain microstructure. This analysis created the DTI outputs, including FA quantitative diffusion maps. Next, the DTI FA images were fed into the tract-based spatial statistics analysis[101], resulting in the skeletonized images. Details of the image acquisition, quality control and processing are described elsewhere (refer to https://biobank.ctsu.ox.ac.uk/crystal/crystal/docs/brain_mri.pdf for the full protocol)[102]. We did not use the imaging-derived phenotypes released by the UK Biobank. Instead, we averaged the skeletonized images of five standard-space tract masks defined by Rojkova et al.[103] by following a processing protocol similar to the ones applied by the UK Biobank and the ENIGMA teams (http://enigma.ini.usc.edu/protocols/dti-protocols). Five left hemisphere white matter tracts that we investigated here are the long segment of the arcuate fasciculus; the SLF subdivisions I, II and III; and the uncinate fasciculus.

**Genome-wide association scanning.** GWASs were performed separately for each of the neuroimaging phenotypes using imputed genotyping data, with PLINK (v1.9)[104]. We made use of categorical and continuous variables controlling for covariates in the GWASs including age, sex, genotype array type and assessment centre. To avoid possible confounding effects related to ancestry, we used the first ten genetic principal components capturing population genetic diversity. These covariates are considered in a pre-residualization step: a multiple linear regression of the endophenotype vector on the covariates is performed, and they are replaced by their corresponding residual. Additionally, a rank-based inverse normalization is performed to ensure that the distributions of endophenotypes are normally distributed.

**Local genetic correlations.** We identified a list of overlapping loci using 2,495 LD blocks covering the whole human genome provided in the LAVA[73] partitioning algorithm GitHub repository (https://github.com/cadeleeuw/lava-partitioning) and 1,609 genome-wide significant ($P < 5 \times 10^{-8}$) SNPs in our $F_{gRI-D}$ summary statistics. This resulted in 18 LD blocks. We then used LAVA to estimate local genetic correlations between $F_{gRI-D}$ and the five aforementioned white matter tracts. LAVA estimates local heritability for each of these 18 LD blocks, and for each considered trait. For the loci that explained a significant proportion (nominally significant SNP heritability estimate, $P < 0.05$) of the total SNP heritability of $F_{gRI-D}$ and white matter tracts, we proceeded to perform bivariate local genetic correlation. This extra step of filtering on the basis of local SNP heritability estimates is not mandatory but recommended[73]. Finally, we obtained local genetic correlation estimates and associated $P$ values, which we FDR corrected for 14 tests.

**Box 1 types of rhythm impairment phenotypes**
Rhythm impairment, which is also described as atypical rhythm, refers to impaired (significantly less accurate) performance on a musical rhythm perception or production task (for example, rhythm and interval discrimination, rhythm or metre processing, beat perception and synchronization and isochronous motor timing or tapping)[16]. It is a broad construct that covers time-based amusia and beat deafness, inaccurate beat synchronization, related impairments in sensitivity to rhythmic patterns and metre of music and inconsistent motor timing[105,106]. The population prevalence of rhythm impairments is estimated to be between 3.0% and 6.5% (ref. 17). The ARRH claims that rhythm impairment is comorbid with developmental language and speech disorders. While the underlying mechanisms of rhythm impairment are largely unknown, ARRH suggests a shared neurobiological and genetic ground for these traits[16].

**Reporting summary**
Further information on research design is available in the Nature Portfolio Reporting Summary linked to this article.

## Data availability
The full GWAS summary statistics from the original 23andMe discovery studies set have been made available through 23andMe to qualified researchers under an agreement with 23andMe that protects the privacy of the 23andMe participants. Datasets will be made available at no cost for academic use. Please visit https://research.23andme.com/collaborate/#dataset-access/ for more information and to apply to access the data. Participants provided informed consent and volunteered to participate in the research online, under a protocol approved by the external AAHRPP-accredited institutional review board, Ethical and Independent Review Services. As of 2022, Ethical and Independent Review Services is part of Salus institutional review board (https://www.versiticlinicaltrials.org/salusirb). The primary neuroimaging genetics data used in this study are available via the UK Biobank website www.ukbiobank.ac.uk. The GWAS summary statistics of FA measures of five white matter tracts, which were derived from the UK Biobank brain imaging dataset, are publicly available at the MPI Archive (accession link: https://hdl.handle.net/1839/d99a85d0-537f-46a2-af19-ee5310311ec8). Genome annotation of the human genome assembly (hg19) was downloaded from the NCBI database (https://www.ncbi.nlm.nih.gov/datasets/gene/GCF_000001405.25/). Source data are provided with this paper.

## Code availability
All scripts used for analyses are publicly available via the GitHub repository at https://github.com/galagoz/pleiotropyevo. This study used openly available software, specifically PLINK (http://zzz.bwh.harvard.edu/plink/) and S-PrediXcan (https://github.com/hakyimlab/MetaXcan). JTI-TWAS prediction models trained on GTEx v8 are available at the PredictDB website (http://predictdb.org and https://github.com/gamazonlab/MR-JTI/tree/master). The human frontal lobe probabilistic atlas used is available at https://storage.googleapis.com/bcblabweb/open_data.html.

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

## Acknowledgements

This project was supported in part by funding from the National Institute on Deafness and Other Communication Disorders, the Office of Behavioural and Social Sciences Research, and the Office of the Director of the National Institutes of Health under Award Numbers R01DC016977, K18DC017383 and DP2HD098859. G.A., E.E., G.B. and S.E.F. are supported by the Max Planck Society. G.B. is also supported by the German Federal Ministry of Education and Research. The funders had no role in study design, data collection and analysis, the decision to publish or the preparation of the manuscript. S.E.F. is a member of the Centre for Academic Research and Training in Anthropogeny. This research was conducted using the UK Biobank resource under application no. 79683. We thank the research participants and employees of 23andMe for making this work possible. The contact for the 23andMe Research Team can be reached at joyce@23andme.com.

## Author contributions

G.A., E.E., N.J.C., R.L.G. and S.E.F. designed the research. G.A., E.E., Y.M. and G.B. performed the research. G.A., E.E. and Y.M. analysed the data. G.A. wrote the initial draft of the paper. E.E., Y.M., G.B., P.F., M.G.N., M.L., R.L.G. and S.E.F. provided critical feedback and commented on the paper.

## Funding

## Competing interests

P.F. is employed by and holds stock or stock options in 23andMe, Inc. The other authors declare no competing interests.

## Additional information

**Correspondence and requests for materials** should be addressed to Gökberk Alagöz, Simon E. Fisher or Reyna L. Gordon.

¹Language and Genetics Department, Max Planck Institute for Psycholinguistics, Nijmegen, the Netherlands. ²Department of Otolaryngology—Head and Neck Surgery, Vanderbilt University Medical Center, Nashville, TN, USA. ³Vanderbilt Genetics Institute, Vanderbilt University Medical Center, Nashville, TN, USA. ⁴Max Planck School of Cognition, Leipzig, Germany. ⁵23andMe, Inc., Sunnyvale, CA, USA. ⁶Population Health Sciences, Bristol Medical School, University of Bristol, Bristol, UK. ⁷MRC Integrative Epidemiology Unit, University of Bristol, Bristol, UK. ⁸Department of Psychology, University of Edinburgh, Edinburgh, UK. ⁹Donders Institute for Brain, Cognition and Behaviour, Radboud University, Nijmegen, the Netherlands. ¹⁰Vanderbilt Brain Institute, Vanderbilt University, Nashville, TN, USA. ¹¹Department of Hearing & Speech Sciences, Vanderbilt University Medical Center, Nashville, TN, USA. ¹²These authors contributed equally: Simon E. Fisher, Reyna L. Gordon. *A list of authors and their affiliations appears at the end of the paper. ✉e-mail: goekberk.alagoez@mpi.nl; simon.fisher@mpi.nl; reyna.gordon@vanderbilt.edu

**23andMe Research Team**

Stella Aslibekyan[5], Adam Auton[5], Elizabeth Babalola[5], Robert K. Bell[5], Jessica Bielenberg[5], Jonathan Bowes[5], Katarzyna Bryc[5], Ninad S. Chaudhary[5], Daniella Coker[5], Sayantan Das[5], Emily DelloRusso[5], Sarah L. Elson[5], Nicholas Eriksson[5], Teresa Filshtein[5], Pierre Fontanillas[5], Will Freyman[5], Zach Fuller[5], Chris German[5], Julie M. Granka[5], Karl Heilbron[5], Alejandro Hernandez[5], Barry Hicks[5], David A. Hinds[5], Ethan M. Jewett[5], Yunxuan Jiang[5], Katelyn Kukar[5], Alan Kwong[5], Yanyu Liang[5], Keng-Han Lin[5], Bianca A. Llamas[5], Matthew H. McIntyre[5], Steven J. Micheletti[5], Meghan E. Moreno[5], Priyanka Nandakumar[5], Dominique T. Nguyen[5], Jared O'Connell[5], Aaron A. Petrakovitz[5], G. David Poznik[5], Alexandra Reynoso[5], Shubham Saini[5], Morgan Schumacher[5], Leah Selcer[5], Anjali J. Shastri[5], Janie F. Shelton[5], Jingchunzi Shi[5], Suyash Shringarpure[5], Qiaojuan Jane Su[5], Susana A. Tat[5], Vinh Tran[5], Joyce Y. Tung[5], Xin Wang[5], Wei Wang[5], Catherine H. Weldon[5], Peter Wilton[5] & Corinna D. Wong[5]

Simon E. Fisher
Reyna L. Gordon

# Reporting Summary

## Statistics

For all statistical analyses, confirm that the following items are present in the figure legend, table legend, main text, or Methods section.

| n/a | Confirmed | |
|---|---|---|
| ☐ | ☒ | The exact sample size (*n*) for each experimental group/condition, given as a discrete number and unit of measurement |
| ☐ | ☒ | A statement on whether measurements were taken from distinct samples or whether the same sample was measured repeatedly |
| ☐ | ☒ | The statistical test(s) used AND whether they are one- or two-sided *Only common tests should be described solely by name; describe more complex techniques in the Methods section.* |
| ☐ | ☒ | A description of all covariates tested |
| ☐ | ☒ | A description of any assumptions or corrections, such as tests of normality and adjustment for multiple comparisons |
| ☐ | ☒ | A full description of the statistical parameters including central tendency (e.g. means) or other basic estimates (e.g. regression coefficient) AND variation (e.g. standard deviation) or associated estimates of uncertainty (e.g. confidence intervals) |
| ☐ | ☒ | For null hypothesis testing, the test statistic (e.g. *F*, *t*, *r*) with confidence intervals, effect sizes, degrees of freedom and *P* value noted *Give P values as exact values whenever suitable.* |
| ☒ | ☐ | For Bayesian analysis, information on the choice of priors and Markov chain Monte Carlo settings |
| ☒ | ☐ | For hierarchical and complex designs, identification of the appropriate level for tests and full reporting of outcomes |
| ☐ | ☒ | Estimates of effect sizes (e.g. Cohen's *d*, Pearson's *r*), indicating how they were calculated |

*Our web collection on statistics for biologists contains articles on many of the points above.*

## Software and code

Policy information about availability of computer code

| Data collection | 23andMe, Inc. and UK Biobank use a custom pipeline for data collection. |
|---|---|
| Data analysis | R (v4.0.3), LD score regression (v1.0.1), PLINK (v1.9), GenomicSEM (v0.0.5c), GWAMA (v1.2.6), CPASSOC (v1), MAGMA (v1.10), GCTB (v2.02), LAVA (v0.1.0). All scripts used in this study are made publicly available at https://github.com/galagoz/pleiotropyevo. 23andMe, Inc. and the UK Biobank use a custom pipeline for data collection. |

For manuscripts utilizing custom algorithms or software that are central to the research but not yet described in published literature, software must be made available to editors and reviewers. We strongly encourage code deposition in a community repository (e.g. GitHub). See the Nature Portfolio guidelines for submitting code & software for further information.

## Data

Policy information about availability of data

All manuscripts must include a data availability statement. This statement should provide the following information, where applicable:
- Accession codes, unique identifiers, or web links for publicly available datasets
- A description of any restrictions on data availability
- For clinical datasets or third party data, please ensure that the statement adheres to our policy

The full GWAS summary statistics from the original 23andMe discovery studies set have been made available through 23andMe to qualified researchers under an agreement with 23andMe that protects the privacy of the 23andMe participants. Datasets will be made available at no cost for academic use. Please visit https://

# Research involving human participants, their data, or biological material

Policy information about studies with human participants or human data. See also policy information about sex, gender (identity/presentation), and sexual orientation and race, ethnicity and racism.

| | |
|---|---|
| Reporting on sex and gender | 23andMe dyslexia GWAS: Number of cases = 51,800 (21,513 male, 30,287 female), Number of controls = 1,087,070 (446,054 male, 641,016 female).<br>23andMe beat-synchronisation GWAS: Total number of cases = 555,660, Number of controls = 51,165 (59% of the total sample size are females).<br>UK Biobank white-matter connectivity GWAS: Total sample size = 31,465 (14,968 male, 16,497 female). |
| Reporting on race, ethnicity, or other socially relevant groupings | Our study sample is limited by the European-ancestry individuals to minimise the population stratification and confounding on GWAS effects. |
| Population characteristics | Population characteristics of dyslexia and beat-synchronisation GWAS study samples are described in the original papers. We used neuroimaging and genotype data from the UK Biobank to perform white-matter connectivity GWASs (N=31,465, mean age=55.21, range between 40 to 70 years old, 16,497 females). The following covariates were used for the white-matter connectivity GWASs: age, sex, genotype array type, assessment centre, and ten genetic principal components capturing population genetic diversity. |
| Recruitment | Participants are customers of 23andMe, so are invited to participate in general research. There is under-representation of low socio-economic position and all participants are over 18 years. The UK Biobank enrolled participants aged 40-69 between 2006 and 2010 for baseline assessments in 22 centres across the UK. The assessment visits comprised interviews and questionnaires covering lifestyles and health conditions, physical measures, biological samples, imaging, and genotyping. |
| Ethics oversight | 23andMe's human subject protocol was reviewed and approved by Ethical & Independent Review Services, a private institutional review board. UK Biobank has received ethical approval from the North West Multi-centre research Ethics Committee (MREC), and informed consent through electronic signature was obtained from study participants. |

Note that full information on the approval of the study protocol must also be provided in the manuscript.

# Field-specific reporting

Please select the one below that is the best fit for your research. If you are not sure, read the appropriate sections before making your selection.

☒ Life sciences ☐ Behavioural & social sciences ☐ Ecological, evolutionary & environmental sciences

For a reference copy of the document with all sections, see nature.com/documents/nr-reporting-summary-flat.pdf

# Life sciences study design

All studies must disclose on these points even when the disclosure is negative.

| | |
|---|---|
| Sample size | Sample sizes (beat-synchronisation N=606,825; dyslexia N=1.138.870) were determined by data availability. This range of sample size has been successfully used in many genome-wide association studies of diverse human phenotypes. |
| Data exclusions | Data exclusion in two original GWAS studies are described in Niarchou et al. (2022) and Doust et al. (2022). For LAVA analysis using diffusion-weighted MRI data, we excluded participants with unusual heterozygosity (>0.19) high missingness (>0.05), sex mismatches between genetically inferred sex and self-reported sex as reported by Bycroft et al. (2018). |
| Replication | Original dyslexia and musical rhythm GWASs (Doust et al., Niarchou et al.) were not divided into discovery and replication samples to maximise the statistical power for genomic-signal discovery. However, we replicated the multivariate GWAS results from Genomic SEM by using two independent multivariate GWAS tools (GWAMA and CPASSOC), and succesfully replicated the shared genetic loci between dyslexia and musical rhythm impairment identified by Genomic SEM (see Supplementary Figure 3).<br><br>Data accession guidelines and accession number in the Data Availability section, scripts provided in the Code Availability section, and the analysis results provided in the Supplementary Information and Supplementary Tables files should be sufficient for other researchers to attempt replication. |
| Randomization | Not relevant, this was an observational study. |
| Blinding | Blinding was not used. Analysts were not blind to phenotypic status. |

# Reporting for specific materials, systems and methods

We require information from authors about some types of materials, experimental systems and methods used in many studies. Here, indicate whether each material, system or method listed is relevant to your study. If you are not sure if a list item applies to your research, read the appropriate section before selecting a response.

## Materials & experimental systems

| n/a | Involved in the study |
|---|---|
| ☒ ☐ | Antibodies |
| ☒ ☐ | Eukaryotic cell lines |
| ☒ ☐ | Palaeontology and archaeology |
| ☒ ☐ | Animals and other organisms |
| ☒ ☐ | Clinical data |
| ☒ ☐ | Dual use research of concern |
| ☒ ☐ | Plants |

## Methods

| n/a | Involved in the study |
|---|---|
| ☒ ☐ | ChIP-seq |
| ☒ ☐ | Flow cytometry |
| ☐ ☒ | MRI-based neuroimaging |

## Plants

| | |
|---|---|
| Seed stocks | *Report on the source of all seed stocks or other plant material used. If applicable, state the seed stock centre and catalogue number. If plant specimens were collected from the field, describe the collection location, date and sampling procedures.* |
| Novel plant genotypes | *Describe the methods by which all novel plant genotypes were produced. This includes those generated by transgenic approaches, gene editing, chemical/radiation-based mutagenesis and hybridization. For transgenic lines, describe the transformation method, the number of independent lines analyzed and the generation upon which experiments were performed. For gene-edited lines, describe the editor used, the endogenous sequence targeted for editing, the targeting guide RNA sequence (if applicable) and how the editor was applied.* |
| Authentication | *Describe any authentication procedures for each seed stock used or novel genotype generated. Describe any experiments used to assess the effect of a mutation and, where applicable, how potential secondary effects (e.g. second site T-DNA insertions, mosiacism, off-target gene editing) were examined.* |

## Magnetic resonance imaging

### Experimental design

| | |
|---|---|
| Design type | No experimental design |
| Design specifications | *Specify the number of blocks, trials or experimental units per session and/or subject, and specify the length of each trial or block (if trials are blocked) and interval between trials.* |
| Behavioral performance measures | *State number and/or type of variables recorded (e.g. correct button press, response time) and what statistics were used to establish that the subjects were performing the task as expected (e.g. mean, range, and/or standard deviation across subjects).* |

### Acquisition

| | |
|---|---|
| Imaging type(s) | Diffusion-weighted MRI |
| Field strength | 3T |
| Sequence & imaging parameters | The UKB Diffusion-weighted MRI images were acquired with five non diffusion-weighted image b = 0 s/mm2, diffusion-weighting of b = 1000, and 2000s/mm2 with 50 directions each. MRI data acquisition details are fully provided here: https://www.nature.com/articles/nn.4393 |
| Area of acquisition | Whole brain |
| Diffusion MRI | ☒ Used    ☐ Not used |
| Parameters | Please see "Sequence & imaging parameters" section above for full details. |

### Preprocessing

| | |
|---|---|
| Preprocessing software | (see https://www.nature.com/articles/nn.4393 and https://www.sciencedirect.com/science/article/pii/S1053811917308613) |
| Normalization | (see https://www.nature.com/articles/nn.4393 and https://www.sciencedirect.com/science/article/pii/S1053811917308613) |

| | |
|---|---|
| Normalization template | (see https://www.nature.com/articles/nn.4393 and https://www.sciencedirect.com/science/article/pii/S1053811917308613) |
| Noise and artifact removal | (see https://www.nature.com/articles/nn.4393 and https://www.sciencedirect.com/science/article/pii/S1053811917308613) |
| Volume censoring | (see https://www.nature.com/articles/nn.4393 and https://www.sciencedirect.com/science/article/pii/S1053811917308613) |

## Statistical modeling & inference

**Model type and settings**
We made use of categorical and continuous variables controlling for covariates in the genome-wide association studies including age, sex, genotype array type, and assessment centre. To avoid possible confounding effects related to ancestry, we used the first ten genetic principal components capturing population genetic diversity.

**Effect(s) tested**
Genetic effects on white-matter connectivity measures, as described in the paper.

**Specify type of analysis:** ☐ Whole brain ☒ ROI-based ☐ Both

**Anatomical location(s)**
We averaged the fractional anisotropy skeletonised image across a set of five left hemishere white-matter tract defined from a probabilistic atlas (Rojkova et al. 2016).

**Statistic type for inference**
(See Eklund et al. 2016)
Local Analysis of [co]Variant Association (LAVA; see https://www.nature.com/articles/s41588-022-01017-y)

**Correction**
FDR

## Models & analysis

| n/a | Involved in the study |
|---|---|
| ☒ ☐ | Functional and/or effective connectivity |
| ☒ ☐ | Graph analysis |
| ☒ ☐ | Multivariate modeling or predictive analysis |

