## [Peer Review File · Nature Human Behaviour]

The shared genetic architecture and evolution of human language and musical rhythm

Corresponding Author: Mr Gökberk Alagöz

This manuscript has been previously reviewed at another journal. This document only contains reviewer comments, rebuttal and decision letters for versions considered at Nature Human Behaviour.

Version 0:

Decision Letter:

16th November 2023

Dear Mr Alagöz,

Thank you once again for your manuscript, entitled "The shared genetic architecture and evolution of human language and musical rhythm", and for your patience during the peer review process.

Your Article has now been evaluated by 3 referees. You will see from their comments copied below that, although they find your work of potential interest, they have raised quite substantial concerns. In light of these comments, we cannot accept the manuscript for publication, but would be interested in considering a revised version if you are willing and able to fully address reviewer and editorial concerns.

We hope you will find the referees' comments useful as you decide how to proceed. If you wish to submit a substantially revised manuscript, please bear in mind that we will be reluctant to approach the referees again in the absence of major revisions. We are committed to providing a fair and constructive peer-review process. Do not hesitate to contact us if there are specific requests from the reviewers that you believe are technically impossible or unlikely to yield a meaningful outcome.

One of the major concerns across the reviewer reports was the strength of the phenotype measurement and self report. Reviewer 3 suggests that additional evidence (including epigenetic) is needed to support your results, and we ask that you take these reviewer suggestions seriously in your revisions.

If you wish to submit a suitably revised manuscript, we would hope to receive it within 4 months. I would be grateful if you could contact us as soon as possible if you foresee difficulties with meeting this target resubmission date.

- Include a "Response to the editors and reviewers" document detailing, point-by-point, how you addressed each editor and referee comment. If no action was taken to address a point, you must provide a compelling argument. When formatting this document, please respond to each reviewer comment individually, including the full text of the reviewer comment verbatim followed by your response to the individual point. This response will be used by the editors to evaluate your revision and sent back to the reviewers along with the revised manuscript.
- Highlight all changes made to your manuscript or provide us with a version that tracks changes.

Link Redacted

Thank you for the opportunity to review your work. Please do not hesitate to contact me if you have any questions or would like to discuss the required revisions further.

Sincerely,

[REDACTED]

Reviewer expertise:

Reviewer #1: beat perception, cogn science

Reviewer #2: speech genetics

Reviewer #3: genetics of musical traits

REVIEWER COMMENTS:

Reviewer #1:

Remarks to the Author:

Relations between music and language processing are an enduring topic interest and debate in the study of the mind, with relevance for basic cognitive science, evolutionary studies, and for the design of clinical applications that use music-based interventions to help individuals with language disorders. As the authors of the current paper note, numerous behavioral studies have found associations between musical rhythmic abilities and certain language skills (e.g., phonological awareness), "implicating potentially shared underlying neural and genetic architectures." These phenotypic associations were recently reviewed by Nayak et al. 2022 in *Neurobiology of Language*, and help motivate a hypothesis positing that individuals with impaired rhythmic abilities are at higher risk of certain developmental speech/language difficulties (the "Atypical Rhythm Risk Hypothesis" or ARRH, Ladanyi et al. 2020).

With this background, the current study breaks new ground and uses genetic data to study music-language relations in the brain. The study takes advantage of recent large-scale GWAS studies of beat synchronization and of dyslexia to examine the extent to which the genetic architecture of musical rhythm and certain aspects of language processing are shared. Among other findings the authors find negative genetic correlations between beat sync abilities and dyslexia, and positive genetic correlations between beat sync abilities and several language abilities: word and nonword reading, phoneme awareness, nonword repetition and spelling. Importantly, the authors found small, nonsignificant genetic correlations between beat sync abilities and non-verbal IQ or overall school performance (Figure 1), indicating that "genetic sharing between rhythm and dyslexia is not driven by general cognition."

Beyond this, the paper has numerous other analyses, including looking at genetic correlations between rhythm impairment and dyslexia, and testing common vs. independent pathway models to tease apart shared genetic substrates of these impairments vs. distinct genetic substrates of each. The paper also looks at cell types implicated in the shared genetic substrates, and conducts evolutionary analyses to see when rhythmic-language genetic overlap emerged in the human lineage. One interesting finding of these additional analyses is that some of the shared genetic substrate for rhythm and language impairment may impact oligodendrocytes and thus myelination and white matter connections that overlap between rhythmic processing and reading-related aspect of language. Consistent with this idea, the authors also found a correlation between shared genetic substrates of rhythm and language impairments and fractional anisotropy (FA) in a long-distance white matter pathway linking parietal and frontal lobes (branch 1 of the superior longitudinal fasciculus, or SLF-1). Another interesting finding (resulting from the evolutionary analyses) is that a SNP identified as being part of a shared genetic substrate of rhythm impairment and dyslexia is in a highly conserved gene (DLAT) whose common polymorphism dates to around the origin of *Homo sapiens*.

Stepping back to the bigger picture, the authors conclude that they "showed robust genetic correlations between rhythm and a number of reading and language-related traits, supporting ARRH." In other words, the paper finds significant associations between the biological foundations of specific aspects of music and language. I suspect this finding will attract a great deal of interest given current debates within cognitive neuroscience over the extent to which music and language processing overlap in the brain, and given growing interest in leveraging music-language relations to improve diagnosis and clinical interventions for those with language disorders.

As I am not a geneticist I cannot evaluate the methods used in the genetic analyses, and will instead focus on things that can help the authors increase the paper's appeal to cognitive scientists. This will likely be the major audience for this work since the paper is submitted to *Nature Human Behaviour*, not to a genetics journal. To maximally engage this audience's interest, I think the authors need to improve the opening of the paper.

The opening sentence of the paper currently states "The human brain has evolved intricate neural circuitry to process complex communicative signals and behaviours, including speech and music..." While there is broad consensus that human brains have evolved specialized circuits for speech processing, there is considerable debate over whether this is also true for music. Many cognitive scientists believe music processing relies on brain circuits that evolved to serve other functions, with no subsequent evolutionary fine-tuning for music (although there can be ontogenetic specialization of circuits based on experience-dependent plasticity). Thus in order not to alienate such skeptics and step into this quagmire in the opening sentence of the paper, I suggest the authors drop the word "evolved" in this sentence and simply say "The human brain has intricate neural circuitry..."

The second sentence of the introduction states "Individual differences in rhythm-related skills (e.g., beat synchronisation, rhythm perception and production, metrical perception) are correlated with variability in a range of language-related skills (e.g., word recognition, spelling, phonological awareness), implicating potentially shared underlying neural and genetic architectures." The

sentence cites Nayak 2022 in support of this assertion. Given that many readers of the current will likely have not read the Nayak review or have followed the music-language literature over the past years, readers may be puzzled to learn that seemingly unrelated abilities like beat synchronization and phonological awareness would be related. They may wonder “how much evidence is there for this?” and “why, from a cognitive or neurobiological perspective, would such associations exist?” and “is this found only in children or also in adults?” etc. Thus in order to get readers on board with the idea that such associations are real and that there are theoretical frameworks which can help account for them, I suggest that the authors add another two concise paragraphs at this point (perhaps in the form of a text box?).

The first new paragraph could clarify what kinds of tasks have been used to establish correlations between rhythm and language skills. The second sentence of the introduction refers to “rhythm perception and production” and “metrical perception”, but these are rather vague terms... what exactly do such tasks ask of participants? Similarly, among the language tasks, what kinds of tasks are being referred to (“word recognition” and “spelling tasks” are rather vague). This paragraph could draw on the Nayak review, e.g., giving some examples of tasks from the studies with larger sample sizes in Fig 5. of that paper. It might be good to mention or visualize how many studies have found such associations, so readers are aware that the evidence for phenotypic associations is not just based on a few studies.

The second new paragraph could point readers to papers that offer reasons for why links between nonlinguistic rhythmic processing and certain language skills would exist in the first place. The current paper cites a 2021 review by Patel hypothesizing that beat processing and vocal learning rely on overlapping brain pathways in a dorsal auditory stream, which provides one reason why rhythm and phonological skills would be related. In addition to that paper, there are other theoretical papers linking rhythmic and language processing, e.g., Tierney, A., & Kraus, N. (2014). Auditory-motor entrainment and phonological skills: precise auditory timing hypothesis (PATH). *Frontiers in Human Neuroscience*, 8, 949. The authors may also want to read/cite Asano, R., Boeckx, C., & Fujita, K. (2022). Moving beyond domain-specific versus domain-general options in cognitive neuroscience. *cortex*, 154, 259-268, and/or theoretical papers that are relevant here.

Having these two new paragraphs in the introduction would likely make readers more receptive to the author’s statement that: “In particular, individuals with rhythm impairment have been suggested to show higher predisposition to language-related difficulties such as dyslexia and developmental language disorder (Atypical Rhythm Risk Hypothesis, ARRH).” Since the ARRH plays an important role in this paper (as evident in the quotes below), it would be good to first mention it after giving some of the empirical and theoretical background on links between rhythm and language processing. This will likely increase interest the readers’ interest in the hypothesis.

ARRH quotes:

“Thus, rhythm is genetically correlated not only with dyslexia, but also multiple language-related phenotypes including word and non-word reading, non-word repetition, phoneme awareness, having better language skills than mathematics, and language resting-state functional connectivity ($|rg|$ median=0.184, range=0.004-0.376), providing empirical genetic evidence for the ARRH.”

“We then performed a mvGWAS on the rhythm impairment and dyslexia GWASs to probe the validity of ARRH at the genetic level...”

“Consistent with the ARRH hypothesis, the directionality of genetic correlations suggest that decreased rhythm impairment/dyslexia risk may be associated with resilience to certain neuropsychiatric disorders.”

(Note that in the above quote, “ARRH hypothesis” is redundant as the H in ARRH stands for hypothesis.)

“In summary, we showed robust genetic correlations between rhythm and a number of reading and language-related traits, supporting ARRH.”

--

Other smaller issues:

In the introduction the authors state: “We hypothesise that identifying the shared genetic architecture between rhythm- and language-related disorders, and probing the evolutionary past of the implicated genomic regions, can help reveal neural and biological characteristics of our species which made rhythm and language an asset to human development and behaviour.” Cognitive science readers may be puzzled by this sentence. The sentence is framed as a hypothesis but it seems more like a statement of belief, not a hypothesis that generates testable predictions. Thus I suggest changing “hypothesize” to “believe”, and staying focused on the ARRH as the paper’s primary hypothesis.

Lines 201-209 refer to studying “evolutionary events” and refers to genes identified in four evolutionary annotations. How far back in time does each of these annotations go (e.g., ancient selective sweep sites)?

Lines 234-236 say “SLF-I is the dorsal division of SLF connecting the superior parietal and superior frontal lobes, with functional links to musical rhythm.” The paper cited at the end of this sentence (about SLF-1 and musical rhythm) is Oechslin et al. (2010). “The plasticity of the superior longitudinal fasciculus as a function of musical expertise: a diffusion tensor imaging study. *Frontiers in Human Neuroscience*, 3, 1076.” That paper examines absolute pitch, not rhythm, and does not dissect the SLF into component tracts, so it does not support a link between SLF-1 and rhythmic processing. In case the authors are interested, there are studies that find links between rhythmic abilities and long-distance white matter structure. Here are two such studies (there are likely others):

Vaquero, L., et al., White-matter structural connectivity predicts short-term melody and rhythm learning in non-musicians.

Neuroimage, 2018. 181: p. 252-262.

Blecher, T., I. Tal, and M. Ben-Shachar, White matter microstructural properties correlate with sensorimotor synchronization abilities. *Neuroimage*, 2016. 138: p. 1-12.

Relatedly, line 246 says the authors found “a local genetic correlation with a tract linked to processing and production of speech and music” and cites Patel 2021 at the end of this sentence as evidence that the tract in question (SLF-1) is involved in speech and music. However, Patel 2021 hypothesized SLF-2 (not SLF-1) was part of the auditory dorsal stream supporting beat processing (see Fig. 3 of that paper).

Lines 239-242 say “The bivariate genomic SEM approach that we developed allowed us to identify genetic overlaps between rhythm impairment and dyslexia, and to present a map of homogeneous and heterogeneous genetic effects, shedding light on patterns of pleiotropy between the two.” This seems like a good place to cite the Musical Abilities, Pleiotropy, Language and Environment (MAPLE) framework introduced in the Nayak 2022 paper. It is somewhat curious that that framework is never mentioned in this paper, since it seems quite relevant to the current work and findings.

Paragraph starting on line 238: Can something be added to this concluding paragraph about the practical implications of these findings? Given that the ARRH is emphasized as a hypothesis underlying the current work, what implications do the current findings have for enhancing early identification of risk for developmental speech/language difficulties?

Reviewer #2:

Remarks to the Author:

Studying the genetic overlap in rhythm and language-related traits is an exciting approach. The authors leveraged their GWAS studies in rhythm and dyslexia to understand the biological overlap in these traits. The outstanding feature of the current study is to pool analytic tools in large datasets to understand the complex traits, as authors have a strong expertise in designing and analyzing large datasets. The authors hypothesized that rhythm-related skill performance correlates with the variability in language-related abilities, suggesting shared biological underpinnings. The study showed the shared genetic markers and evolutionary relationship in language acquisition and musical rhythm. This study provides a foundation for uncovering more genetic markers for the early screening of individuals with language-related disorders. The study used an excellent battery of analytical tools. I have no significant concerns except that little more details can further add to the readers.

The strongest signals to the SNP, rs28576629 in PPP2R3A was validated using additional mvGWAS methods, and the TWAS analysis showed a significant association with several other genes; PPP2R3A expression was among the highest expressed genes. Replication of GWAS signals is a considerable challenge. What do the authors propose to improve the replicability of their findings in other populations and disorders like autism or other behavioral traits that have associated speech and language-related phenotypes?

Reviewer #3:

Remarks to the Author:

The study is based on combining two previously published GWAs, both on commercial GWA studies, one on rhythm (ref 8) and the other on dyslexia (ref 9). In both studies the phenotype definition was based on the announcement of a participant whether he/she can tap according to rhythm or whether he/she has dyslexia. The genetic background of rhythm perception and dyslexia are complex human traits where data is scarce about the effect of family and environmental factors on the trait. Therefore yes/no answer is far away from accurate. Unfortunately the loose definition of the phenotype is unreliable and affects the results.

The conclusions are original. The results may be of interest to people from basic sciences in the field of evolution, music and language.

The validity of the approach and the quality of the presentation are sound. Due to the selected phenotypes it is difficult to evaluate the quality of the data. The authors do not present the limitations of the study other than it has been done using European population.

Suggested improvements: Most likely the participants in the rhythm material were adults whose skills to perceive rhythm is affected by the age and environment where they have grown. What is the role of the environmental effect on the rhythm on their answer?

The same concerns dyslexia. To define the phenotypes in this study, epigenetic studies (methylation array, RNA-studies, miRNA) affecting rhythm and dyslexia are needed. I did not notice any data about the effect of sex on the results.

References: There is a lot of studies about songbirds' vocal learning and its molecular background, f.ex. by the group of ED Jarvis. As songbirds represent a link to humans in evolution, the results of these works should be compared, cited and included in the manuscript.

Clarity and context: p.57. rhythm perception has been important already earlier, not only for present day humans. This part of presentation is lacking the evolutionary view to songbirds, monkeys etc.

p. 173-178 the link to many neuropsychiatric diseases mentioned needs justification. Otherwise it looks like they are just cast to the paper.

Version 1:

Decision Letter:

18th June 2024

Dear Mr Alagöz,

Thank you once again for your revised manuscript, entitled "The shared genetic architecture and evolution of human language and musical rhythm," and for your patience during the re-review process.

Your manuscript has now been evaluated by Reviewers #1 and #2 from the original round of review, as well as a new Reviewer #4 with expertise in Statistical genetics. All reviewer feedback is included at the end of this letter. Although the reviewers found your manuscript to have improved during revision, they also raise some important outstanding concerns. We remain interested in the possibility of publishing your study in Nature Human Behaviour, but would like to consider your response to these outstanding concerns in the form of a revised manuscript before we make a decision on publication.

In particular, we ask that you fully address the concerns raised by Reviewer #1 surrounding the robustness of the neuroanatomical analyses and the lack of information about the spherical deconvolution tractography method.

In sum, we invite you to revise your manuscript taking into account all reviewer and editor comments. We are committed to providing a fair and constructive peer-review process. Do not hesitate to contact us if there are specific requests from the reviewers that you believe are technically impossible or unlikely to yield a meaningful outcome.

We hope to receive your revised manuscript within 4-8 weeks. I would be grateful if you could contact us as soon as possible if you foresee difficulties with meeting this target resubmission date.

- Include a "Response to the editors and reviewers" document detailing, point-by-point, how you addressed each editor and referee comment. If no action was taken to address a point, you must provide a compelling argument. This response will be used by the editors and reviewers to evaluate your revision.
- Highlight all changes made to your manuscript or provide us with a version that tracks changes.

Link Redacted

We look forward to seeing the revised manuscript and thank you for the opportunity to review your work. Please do not hesitate to contact me if you have any questions or would like to discuss these revisions further.

Sincerely,

[REDACTED]

Reviewer expertise:

Reviewer #1: Beat perception

Reviewer #2: Speed genetics

Reviewer #3: Statistical genetics

REVIEWER COMMENTS:

Reviewer #1:

Remarks to the Author:

Note that line numbers below refer to the clean revised ms, not the version with tracked changes.

The authors have been quite responsive to my comments on the original submission, and the paper is much improved. My comments below are aimed at increasing the clarity and impact of the paper.

-

Abstract: The final sentence states: "Overall, we provide empirical evidence of multiple aspects of shared biology linking human rhythm and language that could help resolve long-standing debates over their evolutionary relationships."

If this is to be the "overall" takeaway sentence of the abstract, I suggest changing "human rhythm and language" to "human language and musical rhythm". Putting language first parallels the paper's title, and adding "musical" specifies what type of rhythm is being examined, as "human rhythm" is ambiguous since rhythm is present in speech, locomotion, etc.

(Analogously, in line 72, I suggest changing "language development and rhythm" to "language development and musical rhythm", and in line 295 [first sentence of the Discussion], changing "rhythm" to "musical rhythm".)

-

Line 63 refers to "sporadic beat synchronization", but readers who have not read ref 10 might be confused about what "sporadic" means. I suggest changing this to "predictive and tempo-flexible beat synchronization" (as this mentions key properties of musical beat synchronization) and omitting "sporadic", which readers can learn about by reading ref 10.

-

Lines 68-71 state "Some argue that cognitive systems such as language and music are better understood as different uses of overlapping neural mechanisms¹³, yet the genetic and evolutionary bases of putative shared neural circuits are largely unknown." The Asano et al. 2022 paper cited in this sentence (ref 13) is agnostic about whether the "overlapping neural mechanisms" occur in overlapping or anatomically segregated neural circuits. That is, I think they argue that language and music can engage similar information processing operations even if those operations occur in adjacent (not anatomically overlapping) circuits. Thus to avoid implying that ref 13 is arguing that language and music use some overlapping neural circuits (which would not be a novel argument – see Patel 2003 Nature Neuroscience), I suggest rewording this sentence to say: "Some argue that cognitive systems such as language and music are better understood as different uses of similar information processing mechanisms¹³, yet the genetic and evolutionary bases of putative shared neural circuits are largely unknown."

-

Lines 74-75 say "We believe that identifying potential shared genetic architecture between rhythm- and language-related disorders..." This wording suggests that the paper is examining language disorders in relation to "disorders" of musical rhythm. Calling dyslexia a disorder is already controversial (see Levelt 2001, "Defining dyslexia" Science, 292:1300-1301). Are you sure you want to call a problem clapping in time with a musical beat a disorder? This might seem a bit too pathologizing. Perhaps reword this sentence as "We believe that identifying potential shared genetic architecture between language-related disorders and musical rhythm abilities..."

-

Lines 78-80 say "Importantly, individuals with rhythm impairment have been suggested to show higher predisposition to language-related difficulties such as developmental language disorder and dyslexia (Atypical Rhythm Risk Hypothesis, ARRH)¹⁶." Given how central the ARRH is to this paper, it is important that readers know what you mean by "rhythm impairment." The ARRH paper says "For the purpose of the Atypical Rhythm Risk Hypothesis, we define atypical rhythm as a general construct capturing any/all of the following terms: Impairments in rhythm/beat/meter sensitivity, significantly weaker than normal rhythm ability/skill, poor dynamic attending, beat deafness (Sowinski & Dalla Bella, 2013), or time-based amusia (Peretz & Vuvar, 2017). Atypical rhythm can be classified by poor performance on any implicit or explicit perception or production task of rhythm or timing, such as rhythm discrimination, interval discrimination, rhythm, beat, or meter processing, and synchronization or entrainment."

It would be worth inserting a small table that summarizes this definition of rhythm impairment, so readers are clear on the phenotypes that the ARRH is concerned with.

-

In this revision, the SLF-1 finding is now mentioned in the abstract and is thus highlighted (though oddly, the finding is not mentioned in the Discussion). SLF-1 is not a tract that is typically associated with either language processing or musical rhythm processing, and thus readers familiar with the neurobiology of language or music may treat the finding (and by extension, the entire paper) with skepticism. To protect themselves against such a reaction, the authors should provide more information showing the robustness of their neuroanatomical analyses. Lines 596-600 say: "We extracted the following regions: The left arcuate fasciculus (long, anterior, and posterior segments), the left superior longitudinal fasciculus (I, II, III), and the left uncinate fasciculus for each individual by averaging the FA skeletonised image across a set of five left white-matter tracts defined from a probabilistic atlas¹⁰³."

It is difficult to assess this part of the analysis because although the authors cite a paper which uses spherical deconvolution

tractography, which is presumably what these authors here used to extract the tracts in question, they have not provided enough information to assess the quality of the tractography. It is important to know what software or script they used to run the tractography, with citation; if it was probabilistic or deterministic, and what other parameters were (the number of iterations or streamlines, the curvature setting, whether the tracts were run in native or template space, etc.). The authors also need to indicate what anatomical markers they used, in other words, how they localized and delineated the regions of interest to extract each of the tracts in question. Lastly, and importantly, the tracts in question need to be plotted, either in 2D or 3D space, to be sure they were reconstructed accurately. This plot (along with the other info mentioned above) should be in the supporting information. As a minor comment, I would ask why the authors only reconstructed tracts in the left hemisphere, since the right hemisphere is implicated in auditory processing of non-language sounds which might implicate those areas in beat processing.

Reviewer #2:

Remarks to the Author:

I acknowledge and agree with the author's responses to my comments and revisions to the manuscript. I do not have further queries.

Reviewer #4:

Remarks to the Author:

Having read the original submission, the Reviewers 1-3 comments and authors' answers, plus the revised version of the manuscript, I'm quite convinced of the work in total. The issues raised, for example concerning the validity of the phenotypes such as self-assessed ability to beat to the rhythm and self-reported dyslexia have been well taken into consideration and explained to the reader. Personally, I quite like the idea of utilizing such mass data as collected by eg companies like 23andMe, with self-assessed traits, cause compiling this large numbers of individuals into a study is by other means still quite unaccomplishable or extremely costly. Such data should definitely be utilized as far as possible.

Methodologically, the work is outstanding, with a variety of newest analytical techniques applied. I especially appreciate the methods taking the different evolutionary time frames into consideration, where the natural selection might have taken place in different time depths of primate and human evolution.

The revised manuscript handles well the points made by the reviewers. I warmly recommend the manuscript to be accepted to be published by the Nature Human Behaviour.

Version 2:

Decision Letter:

Our ref: NATHUMBEHAV-23103300B

22nd August 2024

Dear Dr Alagöz,

Thank you for submitting your revised manuscript "The shared genetic architecture and evolution of human language and musical rhythm" (NATHUMBEHAV-23103300B). We find that the paper has improved in revision and will therefore be happy in principle to publish it in Nature Human Behaviour, pending minor revisions to comply with our editorial and formatting guidelines.

We are now performing detailed checks on your paper and will send you a checklist detailing our editorial and formatting requirements within two weeks. Please do not upload the final materials and make any revisions until you receive this additional information from us.

Sincerely,

[REDACTED]

Version 3:

Decision Letter:

Dear Mr Alagöz,

We are pleased to inform you that your Article "The shared genetic architecture and evolution of human language and musical rhythm", has now been accepted for publication in *Nature Human Behaviour*.

Please note that *Nature Human Behaviour* is a Transformative Journal (TJ). Authors may publish their research with us through the traditional subscription access route or make their paper immediately open access through payment of an article-processing charge (APC). Authors will not be required to make a final decision about access to their article until it has been accepted. [Find out more about Transformative Journals](https://www.springernature.com/gp/open-research/transformative-journals)

Authors may need to take specific actions to achieve [compliance with funder and institutional open access mandates](https://www.springernature.com/gp/open-research/funding/policy-compliance-faqs). If your research is supported by a funder that requires immediate open access (e.g. according to [Plan S principles](https://www.springernature.com/gp/open-research/plan-s-compliance)) then you should select the gold OA route, and we will direct you to the compliant route where possible. For authors selecting the subscription publication route, the journal's standard licensing terms will need to be accepted, including [self-archiving policies](https://www.springernature.com/gp/open-research/policies/journal-policies). Those licensing terms will supersede any other terms that the author or any third party may assert apply to any version of the manuscript.

We welcome the submission of potential cover material (including a short caption of around 40 words) related to your manuscript; suggestions should be sent to *Nature Human Behaviour* as electronic files (the image should be 300 dpi at 210 x 297 mm in either TIFF or JPEG format). Please note that such pictures should be selected more for their aesthetic appeal than for their scientific content, and that colour images work better than black and white or grayscale images. Please do not try to design a cover with the *Nature Human Behaviour* logo etc., and please do not submit composites of images related to your work. I am sure you will understand that we cannot make any promise as to whether any of your suggestions might be selected for the cover of the journal.

With best regards,

[REDACTED]

P.S. Click on the following link if you would like to recommend *Nature Human Behaviour* to your librarian <http://www.nature.com/subscriptions/recommend.html#forms>

** Visit the Springer Nature Editorial and Publishing website at http://editorial-jobs.springernature.com?utm_source=ejP_NHumB_email&utm_medium=ejP_NHumB_email&utm_campaign=ejP_NHumB

and-publishing-jobs for more information about our career opportunities. If you have any questions please click here.**

**Response to reviewers' comments for Nature Human Behaviour manuscript
NATHUMBEHAV-23103300-T titled "The shared genetic architecture and evolution of
human language and musical rhythm"**

- Author responses are in blue. Additions to the manuscript are in blue+bold.
-

Reviewer Comments:

Reviewer #1:

Relations between music and language processing are an enduring topic interest and debate in the study of the mind, with relevance for basic cognitive science, evolutionary studies, and for the design of clinical applications that use music-based interventions to help individuals with language disorders. As the authors of the current paper note, numerous behavioral studies have found associations between musical rhythmic abilities and certain language skills (e.g., phonological awareness), "implicating potentially shared underlying neural and genetic architectures." These phenotypic associations were recently reviewed by Nayak et al. 2022 in *Neurobiology of Language*, and help motivate a hypothesis positing that individuals with impaired rhythmic abilities are at higher risk of certain developmental speech/language difficulties (the "Atypical Rhythm Risk Hypothesis" or ARRH, Ladanyi et al. 2020).

With this background, the current study breaks new ground and uses genetic data to study music-language relations in the brain. The study takes advantage of recent large-scale GWAS studies of beat synchronization and of dyslexia to examine the extent to which the genetic architecture of musical rhythm and certain aspects of language processing are shared. Among other findings the authors find negative genetic correlations between beat sync abilities and dyslexia, and positive genetic correlations between beat sync abilities and several language abilities: word and nonword reading, phoneme awareness, nonword repetition and spelling. Importantly, the authors found small, nonsignificant genetic correlations between beat sync abilities and non-verbal IQ or overall school performance (Figure 1), indicating that "genetic sharing between rhythm and dyslexia is not driven by general cognition."

Beyond this, the paper has numerous other analyses, including looking at genetic correlations between rhythm impairment and dyslexia, and testing common vs. independent pathway models to tease apart shared genetic substrates of these impairments vs. distinct genetic

substrates of each. The paper also looks at cell types implicated in the shared genetic substrates, and conducts evolutionary analyses to see when rhythmic-language genetic overlap emerged in the human lineage. One interesting finding of these additional analyses is that some of the shared genetic substrate for rhythm and language impairment may impact oligodendrocytes and thus myelination and white matter connections that overlap between rhythmic processing and reading-related aspect of language. Consistent with this idea, the authors also found a correlation between shared genetic substrates of rhythm and language impairments and fractional anisotropy (FA) in a long-distance white matter pathway linking parietal and frontal lobes (branch 1 of the superior longitudinal fasciculus, or SLF-1). Another interesting finding (resulting from the evolutionary analyses) is that a SNP identified as being part of a shared genetic substrate of rhythm impairment and dyslexia is in a highly conserved gene (DLAT) whose common polymorphism dates to around the origin of *Homo sapiens*.

Stepping back to the bigger picture, the authors conclude that they “showed robust genetic correlations between rhythm and a number of reading and language-related traits, supporting ARRH.” In other words, the paper finds significant associations between the biological foundations of specific aspects of music and language. I suspect this finding will attract a great deal of interest given current debates within cognitive neuroscience over the extent to which music and language processing overlap in the brain, and given growing interest in leveraging music-language relations to improve diagnosis and clinical interventions for those with language disorders.

We thank the Reviewer for the extensive and precise summary of our manuscript.

As I am not a geneticist I cannot evaluate the methods used in the genetic analyses, and will instead focus on things that can help the authors increase the paper’s appeal to cognitive scientists. This will likely be the major audience for this work since the paper is submitted to *Nature Human Behaviour*, not to a genetics journal. To maximally engage this audience’s interest, I think the authors need to improve the opening of the paper.

The opening sentence of the paper currently states “The human brain has evolved intricate neural circuitry to process complex communicative signals and behaviours, including speech and music...” While there is broad consensus that human brains have evolved specialized

circuits for speech processing, there is considerable debate over whether this is also true for music. Many cognitive scientists believe music processing relies on brain circuits that evolved to serve other functions, with no subsequent evolutionary fine-tuning for music (although there can be ontogenetic specialization of circuits based on experience-dependent plasticity). Thus in order not to alienate such skeptics and step into this quagmire in the opening sentence of the paper, I suggest the authors drop the word “evolved” in this sentence and simply say “The human brain has intricate neural circuitry...”

The Reviewer draws our attention to a critical point here. We agree that the lack of a consensus over the (co)evolution of rhythm and language may spark some criticism on this sentence of our manuscript. Thus, following the Reviewer’s suggestion, we dropped the word “evolved”, and updated the first sentence of the Introduction as below:

“The human brain has intricate neural circuitry to process complex communicative signals and behaviours, including speech and music, and the extent of biological overlap between these facets is an important question for the field of neurobiology.” (page 2, lines 34-36)

The second sentence of the introduction states “Individual differences in rhythm-related skills (e.g., beat synchronisation, rhythm perception and production, metrical perception) are correlated with variability in a range of language-related skills (e.g., word recognition, spelling, phonological awareness), implicating potentially shared underlying neural and genetic architectures.” The sentence cites Nayak 2022 in support of this assertion. Given that many readers of the current will likely have not read the Nayak review or have followed the music-language literature over the past years, readers may be puzzled to learn that seemingly unrelated abilities like beat synchronization and phonological awareness would be related. They may wonder “how much evidence is there for this?” and “why, from a cognitive or neurobiological perspective, would such associations exist?” and “is this found only in children or also in adults?” etc. Thus in order to get readers on board with the idea that such associations are real and that there are theoretical frameworks which can help account for them, I suggest that the authors add another two concise paragraphs at this point (perhaps in the form of a text box?).

The first new paragraph could clarify what kinds of tasks have been used to establish correlations between rhythm and language skills. The second sentence of the introduction refers to “rhythm perception and production” and “metrical perception”, but these are rather vague terms... what exactly do such tasks ask of participants? Similarly, among the language tasks, what kinds of tasks are being referred to (“word recognition” and “spelling tasks” are rather vague). This paragraph could draw on the Nayak review, e.g., giving some examples of tasks from the studies with larger sample sizes in Fig 5. of that paper. It might be good to mention or visualize how many studies have found such associations, so readers are aware that the evidence for phenotypic associations is not just based on a few studies.

We thank the Reviewer for these clear pointers to clarify phenotype measurement tasks, which will indeed improve our manuscript’s accessibility to a broader readership. We now added two new paragraphs to the Introduction, and restructured some of the extant paragraphs by using the Reviewer’s comments as a guide. Below is the first new Introduction paragraph, tapping into the phenotypic links between rhythm- and language-related skills, and phenotype measurement task descriptions by using the rhythm-related studies compiled in Nayak et al. Fig. 5:

“Individual differences in rhythm-related skills are correlated with variability in language-related skills, implicating potentially shared underlying neural and genetic architectures¹. Previous research on the relationship between rhythm- and language-related skills used a wide range of task-based tests to measure aspects of rhythm (e.g., beat synchronisation, rhythm perception and production, metrical perception) and spoken and written language abilities (e.g., word recognition, spelling, phonological awareness). For instance, rhythm perception skills can be measured by quantifying participants’ ability to discriminate differences in durations of adjacent tones in melodies, whereas beat synchronisation skills can be quantified based on participants’ success in tapping along to a metronome² or to the beat (pulse) of musical excerpts³. Similarly, language-related skills such as word reading ability can be measured as the ability to sound out words quickly and accurately in a limited amount of time⁴, and spelling skills can be measured by testing participants’ abilities to correctly spell out a number of words read aloud by the tester⁵. Importantly, even though language and rhythm measurement tasks involve different signals and stimuli, and capture different skills, studies of individual differences often show phenotypic correlations between the

different traits⁶. Nayak et al.¹ compiled information from 25 studies that identified significant positive phenotypic correlations between rhythm and language-related skills, synthesizing findings on a total of 397 children and 606 adults. Consistent associations have been found between rhythm perception, beat synchronisation, and language-related skills including speech perception, word reading and grammatical skills at various phases in the lifespan from preschool age through to later adulthood. Despite these lines of evidence showing phenotypic associations between non-linguistic rhythmic processing and language skills, the shared neurobiological, genetic and evolutionary grounds of these traits remain to be discovered.” (page 2, lines 36-59)

The second new paragraph could point readers to papers that offer reasons for why links between nonlinguistic rhythmic processing and certain language skills would exist in the first place. The current paper cites a 2021 review by Patel hypothesizing that beat processing and vocal learning rely on overlapping brain pathways in a dorsal auditory stream, which provides one reason why rhythm and phonological skills would be related. In addition to that paper, there are other theoretical papers linking rhythmic and language processing, e.g., Tierney, A., & Kraus, N. (2014). Auditory-motor entrainment and phonological skills: precise auditory timing hypothesis (PATH). *Frontiers in Human Neuroscience*, 8, 949. The authors may also want to read/cite Asano, R., Boeckx, C., & Fujita, K. (2022). Moving beyond domain-specific versus domain-general options in cognitive neuroscience. *cortex*, 154, 259-268, and/or theoretical papers that are relevant here.

In the second new introductory paragraph, as suggested by the Reviewer, we provided some perspective on the theoretical grounds of rhythm-language associations to improve the accessibility of the manuscript to those who are not familiar with the related literature:

“Various theoretical frameworks^{7,8,9}, such as the revised vocal learning hypothesis¹⁰, provide overarching perspectives on how rhythm and multiple facets of human communication might relate in a neurodevelopmental and evolutionary context. According to the revised vocal learning hypothesis, human vocal learning ability is a preadaptation for sporadic beat synchronisation, and beat processing and vocal learning rely on overlapping neural circuits. This view is in line with neural reuse theories, such as neuronal recycling¹¹ and massive redeployment hypotheses¹², which suggest overlapping neurobiological circuits for language- and rhythm-related skills.

Neural reuse theories claim that cultural innovations like reading invade evolutionarily older brain substrates via the reallocation of an existing neural circuit to a new behaviour. Some argue that cognitive systems such as language and music are better understood as different uses of overlapping neural mechanisms¹³, yet the genetic and evolutionary bases of putative shared neural circuits are largely unknown.” (pages 2-3, lines 60-71)

Having these two new paragraphs in the introduction would likely make readers more receptive to the author’s statement that: “In particular, individuals with rhythm impairment have been suggested to show higher predisposition to language-related difficulties such as dyslexia and developmental language disorder (Atypical Rhythm Risk Hypothesis, ARRH).” Since the ARRH plays an important role in this paper (as evident in the quotes below), it would be good to first mention it after giving some of the empirical and theoretical background on links between rhythm and language processing. This will likely increase interest the readers’ interest in the hypothesis.

We now moved this hypothesis sentence under the two newly added paragraphs. We believe that the addition of two new paragraphs and restructuring of the Introduction section substantially improved our manuscript. We thank the Reviewer for the valuable suggestions and guidance to modify our Introduction suitably for cognitive science readership.

ARRH quotes:

“Thus, rhythm is genetically correlated not only with dyslexia, but also multiple language-related phenotypes including word and non-word reading, non-word repetition, phoneme awareness, having better language skills than mathematics, and language resting-state functional connectivity ($|r_g|$ median=0.184, range=0.004-0.376), providing empirical genetic evidence for the ARRH.”

“We then performed a mvGWAS on the rhythm impairment and dyslexia GWASs to probe the validity of ARRH at the genetic level...”

“Consistent with the ARRH hypothesis, the directionality of genetic correlations suggest that decreased rhythm impairment/dyslexia risk may be associated with resilience to certain neuropsychiatric disorders.”

(Note that in the above quote, “ARRH hypothesis” is redundant as the H in ARRH stands for hypothesis.)

“In summary, we showed robust genetic correlations between rhythm and a number of reading and language-related traits, supporting ARRH.”

Thanks very much for compiling the ARRH quotes in our manuscript, which indicates the centrality of this hypothesis to our work. Also, we thank the Reviewer for pointing out the redundant use of the word “hypothesis” after “ARRH”, which we now removed.

Other smaller issues:

In the introduction the authors state: “We hypothesise that identifying the shared genetic architecture between rhythm- and language-related disorders, and probing the evolutionary past of the implicated genomic regions, can help reveal neural and biological characteristics of our species which made rhythm and language an asset to human development and behaviour.” Cognitive science readers may be puzzled by this sentence. The sentence is framed as a hypothesis but it seems more like a statement of belief, not a hypothesis that generates testable predictions. Thus I suggest changing “hypothesize” to “believe”, and staying focused on the ARRH as the paper’s primary hypothesis.

We thank the Reviewer for pointing this out. Indeed, this sentence states our overall research goal rather than being a testable hypothesis, so we replaced the word “hypothesize” with “believe” now. We agree that the ARRH stands out more clearly as the main hypothesis of our study now.

Lines 201-209 refer to studying “evolutionary events” and refers to genes identified in four evolutionary annotations. How far back in time does each of these annotations go (e.g., ancient selective sweep sites)?

Each gene-set and its corresponding evolutionary genetic annotation are linked to different time windows of evolution: Differentially Methylated Regions between Anatomically Modern Humans and archaic humans to ~650,000-35,000, Ancient Selective Sweeps sites to 650,000-100,000, Human Accelerated Regions to ~8 million, and Differentially Methylated Regions between Anatomically Modern Humans and chimpanzees to ~8 million years ago.

We now clarified this in the manuscript as follows, and included gene-set specific time windows in a new column named “Notes” in Supplementary Table 21:

“Next, we performed MAGMA gene-set analysis⁶¹ to investigate links between genes that are co-located with various evolutionary annotations, spanning a timescale from ~8 million to ~35,000 years ago, which were not testable via partitioned heritability analysis due to low SNP-coverage (see Methods).” (page 8-9, lines 243-246)

Lines 234-236 say “SLF-I is the dorsal division of SLF connecting the superior parietal and superior frontal lobes, with functional links to musical rhythm.” The paper cited at the end of this sentence (about SLF-1 and musical rhythm) is Oechslin et al. (2010). “The plasticity of the superior longitudinal fasciculus as a function of musical expertise: a diffusion tensor imaging study. *Frontiers in Human Neuroscience*, 3, 1076.” That paper examines absolute pitch, not rhythm, and does not dissect the SLF into component tracts, so it does not support a link between SLF-1 and rhythmic processing. In case the authors are interested, there are studies that find links between rhythmic abilities and long-distance white matter structure. Here are two such studies (there are likely others):

Vaquero, L., et al., White-matter structural connectivity predicts short-term melody and rhythm learning in non-musicians. *Neuroimage*, 2018. 181: p. 252-262.

Blecher, T., I. Tal, and M. Ben-Shachar, White matter microstructural properties correlate with sensorimotor synchronization abilities. *Neuroimage*, 2016. 138: p. 1-12.

Many thanks to the Reviewer for carefully pointing out these erroneous citations in our manuscript, and for providing alternative sources. Indeed, we noticed that Oechslin et al.

(2010) does not dissect the SLF into component tracts, so we replaced this citation with Blecher et al. (2016) and Vaquero et al. (2018) as the Reviewer suggested.

“SLF-I is the dorsal division of SLF connecting the superior parietal and superior frontal lobes⁷⁵. Even though SLF was shown to have functional links to musical rhythm^{76,77}, the SLF-I subdivision is mostly involved in the regulation of motor behaviour^{78,79}. This finding is partially consistent with the hypothesized role of the dorsal stream in supporting co-evolution of phonological processing and beat synchronisation⁴.” (page 10, lines 288-293)

Relatedly, line 246 says the authors found “a local genetic correlation with a tract linked to processing and production of speech and music” and cites Patel 2021 at the end of this sentence as evidence that the tract in question (SLF-1) is involved in speech and music. However, Patel 2021 hypothesized SLF-2 (not SLF-1) was part of the auditory dorsal stream supporting beat processing (see Fig. 3 of that paper).

This is indeed another error, we apologise for the confusion. We now replaced this citation and changed our interpretation, as SLF-I is functionally less language-related, and is not connected to Broca’s area (doi.org/10.1016/j.bandl.2013.03.001):

“... and a local genetic correlation with a tract linked to regulation of higher aspects of motor behaviour⁸²” (page 11, lines 320-3321)

Lines 239-242 say “The bivariate genomic SEM approach that we developed allowed us to identify genetic overlaps between rhythm impairment and dyslexia, and to present a map of homogeneous and heterogeneous genetic effects, shedding light on patterns of pleiotropy between the two.” This seems like a good place to cite the Musical Abilities, Pleiotropy, Language and Environment (MAPLE) framework introduced in the Nayak 2022 paper. It is somewhat curious that that framework is never mentioned in this paper, since it seems quite relevant to the current work and findings.

We used the MAPLE framework as a basis for the conceptualisation of our study, and already cited Nayak et al. (2022) in our Introduction. Yet, we indeed have not mentioned or cited the MAPLE framework in the Discussion. Now, we cited Nayak et al. (2022) after the

aforementioned sentence once again, which further clarifies the importance of this framework for our study.

Paragraph starting on line 238: Can something be added to this concluding paragraph about the practical implications of these findings? Given that the ARRH is emphasized as a hypothesis underlying the current work, what implications do the current findings have for enhancing early identification of risk for developmental speech/language difficulties?

We believe that our findings might constitute an important first step towards understanding the genetics of developmental speech and language problems, and its links with musical rhythm skills. In theory, summary statistics derived from the common genetic factor results can be used to estimate a “shared” polygenic score for proneness to dyslexia and rhythm impairment using standard polygenic score estimation methods. However, this should be approached with caution and, like other polygenic scores, its true predictive value in a clinical or early risk detection context remains to be investigated. We now discuss this potential future use of our mvGWAS results in Discussion:

“Among 18 genome-wide significant loci associated with the common factor of dyslexia and rhythm impairment, the strongest mvGWAS signal comes from a locus tagged by the SNP rs28576629 that is mapped to *PPP2R3A*, a gene implicated in the negative control of cell growth and division, suggesting a putative role for this gene in dyslexia and rhythm impairment prevalence⁸⁰. Given that we validated our common factor results using two additional mvGWAS methods, we believe that the shared genetic factor that we captured represents a solid first glimpse into the shared genetics of dyslexia and rhythm impairment. Results of this kind might potentially contribute, together with information on other risk factors, towards improved diagnostics of individuals’ propensity for reading and rhythm-related problems, to enable special educational support. However, given the highly polygenic and environment-dependent nature of behavioural traits, the early risk identification power of our results remains to be explored.⁸¹” (pages 11, lines 305-316)

Reviewer #2:

Studying the genetic overlap in rhythm and language-related traits is an exciting approach. The authors leveraged their GWAS studies in rhythm and dyslexia to understand the biological overlap in these traits. The outstanding feature of the current study is to pool analytic tools in large datasets to understand the complex traits, as authors have a strong expertise in designing and analyzing large datasets. The authors hypothesized that rhythm-related skill performance correlates with the variability in language-related abilities, suggesting shared biological underpinnings. The study showed the shared genetic markers and evolutionary relationship in language acquisition and musical rhythm. This study provides a foundation for uncovering more genetic markers for the early screening of individuals with language-related disorders. The study used an excellent battery of analytical tools. I have no significant concerns except that little more details can further add to the readers.

The strongest signals to the SNP, rs28576629 in PPP2R3A was validated using additional mvGWAS methods, and the TWAS analysis showed a significant association with several other genes; PPP2R3A expression was among the highest expressed genes. Replication of GWAS signals is a considerable challenge. What do the authors propose to improve the replicability of their findings in other populations and disorders like autism or other behavioral traits that have associated speech and language-related phenotypes?

We thank the Reviewer for the summary and positive evaluation of our work. Indeed, the replicability of GWAS signals and post-GWAS analysis results in different ancestral and/or clinical populations (i.e. transferability) is a challenge of the field. There are multiple sources of this problem, including statistical (e.g., sample size differences between cohorts and ancestries, sampling bias), genetic (e.g., differences in linkage disequilibrium and minor allele frequency, and gene-gene interactions) and environmental (e.g., assortative mating, dynastic effects, socioeconomic status, population stratification, and gene-environment interactions) factors (doi.org/10.1016/j.ajhg.2022.12.011). Replicability is particularly challenging for psychiatric and behavioural traits, as the Reviewer rightly pointed out, due to higher environmental and cultural confounds on such traits, which is also reflected in the poor transferability of polygenic scores across genetic ancestries compared to other types of complex traits (doi.org/10.1016/j.biopsych.2019.06.007).

Importantly, the complex trait genetics field has been working hard over the last decade to address these issues. Thanks to the advent of large-scale biobanks, and meta-analysis

consortiums, GWAS sample sizes have reached to the order of millions of participants for some traits (e.g. height, dyslexia) in European ancestry cohorts, providing an unprecedented statistical power for variant-trait association detection. Unfortunately, GWASs of non-European cohorts are quite far from such sample sizes, which obstructs capturing the “species-wide” genetic architecture of a trait, independent from population- and environment-specific signals. To address this, we think that European and North American research institutes should collaborate more with non-European research institutes, and contribute to the ongoing local efforts of large-scale GWASs in under-represented populations. Also, new resources such as NIH All of Us are promising in terms of providing larger and more diverse study samples, revealing a bigger picture of human genetic diversity. We also actively develop and promote the development of new GWAS methods or extensions on existing tools (such as the bivariate GenomicSEM approach that we developed as part of this project) to address the shortcomings of existing GWAS methods. As the GWAS sample sizes and methods improve, we believe that a clearer and more complete picture of complex trait genetics will emerge.

We now added a new paragraph to the Discussion where we summarise replicability-related issues in the GWAS field, and propose potential improvements:

“Our study has several limitations. First, the original dyslexia and rhythm GWASs were performed on European ancestry samples, due to data availability reasons. The lack of large-scale GWASs in non-European populations and the strong sampling bias in large cohorts towards European ancestry individuals hinder a global picture of the genetic architecture of these traits, weaken the replicability of GWAS findings in diverse populations⁸⁵, and limit the interpretability of post-GWAS evolutionary analysis results. This is especially true for behavioural and psychiatric traits that are more prone to be affected by cultural, socioeconomic and environmental factors, which is also reflected in the weaker transferability of polygenic scores across ancestries for such traits⁸⁶. We believe that important steps to solve this will include encouraging and contributing to the generation of large-scale databases with genotype/phenotype data of individuals from diverse genetic ancestries, disentangling the unique gene-environment interactions in other populations, developing GWAS methods and study designs to more carefully take potential confounders into account, and improving the reliability of behavioural phenotype measurement techniques.” (pages 12-13, lines 359-372)

Reviewer #3:

The study is based on combining two previously published GWAs, both on commercial GWA studies, one on rhythm (ref 8) and the other on dyslexia (ref 9). In both studies the phenotype definition was based on the announcement of a participant whether he/she can tap according to rhythm or whether he/she has dyslexia. The genetic background of rhythm perception and dyslexia are complex human traits where data is scarce about the effect of family and environmental factors on the trait. Therefore yes/no answer is far away from accurate. Unfortunately the loose definition of the phenotype is unreliable and affects the results.

The conclusions are original. The results may be of interest to people from basic sciences in the field of evolution, music and language.

We thank the Reviewer for the objective evaluation of our work and for constructive criticisms. We agree that the self-report based binary phenotype descriptions in the original GWASs are not the most ideal phenotype measurement methods, and environmental factors such as socioeconomic status, assortative mating, dynastic effects and culture can certainly impact the prevalence and diagnosis rates of language/reading- and rhythm-related problems. Yet, there is converging evidence showing that the original GWASs largely capture true genetic factors shaping dyslexia and rhythm skills, and we do not think environmental confounders are substantially masking the genetic effects in these very large GWASs (with N's of up to 1.1 million people). Our work, making pragmatic use of more easily assessed phenotypic indicators to substantially enhance power for large-scale studies of genetic components is fully in line with the state-of-the-art in the field of complex traits. Using the Reviewer's valuable suggestions as a guide, here we present multiple lines of independent evidence supporting the validity of our GWAS phenotypes. We discuss each highlighted issue in the Discussion, as noted:

Evidence 1 – Diagnosis-based dyslexia self-report question uncovers highly stable genetic architecture across the lifespan in a cohort spanning ages 18-110 years:

“Second, the self-report based phenotype descriptions in the original GWASs are not ideal measures, but rather represent robust validated proxies that uniquely enable

scaling up of data collection to very large cohorts. There is a compromise between the practical convenience of self-report based data collection from hundreds of thousands of individuals, which is extremely challenging using task-based measurements, and introducing self assessment-related uncertainties to the data. We note that the phenotype in the dyslexia GWAS was not simply self assessment but rather self report of having received a positive dyslexia diagnosis, and that the genetic architecture was found to be stable across the lifespan, with a genetic correlation of 0.97 between younger (<55 years) and older subgroups (>55 years) of participants¹⁸.” (page 13, lines 372-381)

Evidence 2 - GWAS rhythm phenotype has been validated as a reliable indicator of tested abilities in independent studies, and polygenic scores correlate with performance on an external rhythm discrimination test:

“On the other hand, the validity of the rhythm GWAS phenotype is supported by the phenotype validation studies in independent samples¹⁷, while polygenic scores trained on the rhythm summary statistics have been shown to correlate with a rhythm discrimination test^{87,88}, further supporting the view that genetic signals associated with self-reported beat synchronisation ability indeed relate to rhythm-related skills.” (page 13, lines 381-385)

Evidence 3 – The self-report dyslexia and rhythm impairment phenotypes show robust significant genetic correlations with an independent set of language-related traits that were quantitatively measured through psychometric testing in up to 34,000 individuals:

“Moreover, the genetic correlations between dyslexia, rhythm impairment, $F_{\text{gRI-D}}$, and other speech and language-related phenotypes suggest that the original GWASs largely capture relevant genetic factors.” (page 13, lines 385-387)

Evidence 4 – Significant $F_{\text{gRI-D}}$ SNP-heritability enrichment is observed in neuronal and non-neuronal brain cell types:

“Our SNP-heritability enrichment results in cell-type specific regulatory regions point to multiple brain cell-types for follow-up work, without pinning down a specific neuronal or non-neuronal brain cell-type, indicating a complex cellular composition in the brain supporting rhythm and language. Heritability enrichment signals in brain-specific regulatory regions provide additional evidence that the dyslexia and rhythm

GWASs largely capture the neurodevelopmental aspects of these traits.” (page 11, lines 327-332)

The validity of the approach and the quality of the presentation are sound. Due to the selected phenotypes it is difficult to evaluate the quality of the data. The authors do not present the limitations of the study other than it has been done using European population.

We thank the Reviewer for the positive evaluation of our work, and for pointing out an important shortcoming of the manuscript regarding the presentation of limitations. In addition to the newly added text on the replicability and phenotype description issues (please see above), here we further discuss the limitations of our work (e.g., limitations related to the use of present-day variation for evolutionary studies, the difficulties in interpretability of heritability-based evolutionary analyses). We also propose potential solutions for each issue:

“Fourth, the majority of genetic factors shaping human language and musical rhythm skills are likely fixed in all human populations. Hence, post-GWAS evolutionary analyses, which leverage present-day variation to probe links between genetic variants and evolutionary annotations, are not ideal to study the origins of traits that likely emerged at earlier periods of hominin evolution. Methodological developments in the complex trait evolution field, and the integration of ancient DNA data from older time points of human evolution into polygenic selection analysis methods would greatly help to resolve these challenges in future studies. Finally, relationships between heritability and evolution are intricate. The individual contributions of common genetic variants to heritability are jointly shaped by selection and demography⁸⁹, which limits the evolutionary interpretation of heritability-based methods..” (page 13-14, lines 392-402)

Suggested improvements: Most likely the participants in the rhythm material were adults whose skills to perceive rhythm is affected by the age and environment where they have grown. What is the role of the environmental effect on the rhythm on their answer? The same concerns dyslexia. To define the phenotypes in this study, epigenetic studies (methylation array, RNA-studies, miRNA) affecting rhythm and dyslexia are needed. I did not notice any data about the effect of sex on the results.

References: There is a lot of studies about songbirds' vocal learning and its molecular background, f.ex. by the group of ED Jarvis. As songbirds represent a link to humans in evolution, the results of these works should be compared, cited and included in the manuscript.

The Reviewer is drawing attention to multiple non-genetic factors that can influence rhythm and dyslexia phenotypes, including environmental, epigenetic and sex effects. Our work here employs the GWAS study design, which is widely accepted as one of the most powerful tools for teasing out the molecular genetic architecture underlying a phenotype of interest. As noted above, we agree that the self-report based binary phenotype descriptions in the original GWASs (Niarchou et al., 2022 and Doust et al., 2022) are not the most ideal phenotypic measures; yet, regardless of the precision of phenotypic measurement, elucidation of the impacts of environmental factors on dyslexia and rhythm will not come from GWAS and post-GWAS analyses alone - other approaches from an array of fields are much better suited for addressing those questions. Even in a study with deeply detailed behavioural/clinical phenotypes, it may not be fully possible to unravel the biological consequences of genetic variation and the ways in which other non-genetic factors intercept those biological processes to influence observable phenotypes. Instead, just as in GWAS-based investigations of other traits, our current study can draw knowledge from multiple types of evidence and data sources to begin to piece together the chain of events by which DNA variation influences phenotypic outcomes. Below, we summarise the evidence and the ways in which we have revised the manuscript to address these comments within the scope of GWAS-related analyses, including a new transcriptomics-based analysis to capture some of that biology. We have now integrated these new results and text to the main manuscript and the supplementary material where appropriate.

We acknowledge that factors such as the environment in which the GWAS participants grew up in, their socioeconomic status, sex, and overall health conditions can introduce some bias or error into any phenotyping survey, especially when it comes to behavioural phenotypes. Yet, previous studies estimated small and trivial effects of familial environments, beyond genetic effects, on objective tests of rhythm discrimination abilities. For example, Ullén et al. (2014) (doi.org/10.1016/j.jpaid.2014.01.057) employed a twin design in a large sample of deeply phenotyped twin pairs and estimated only 2% of the total variance in rhythm discrimination ability to be explained by the environment shared between twin siblings (an

effect that did not reach statistical significance). We note that indirect genetic confounding is prevalent when indirect transmission, e.g., shared environmental effects or passive gene-environment correlations, is at play (e.g., educational attainment) (for reference, please see Silventoinen et al., 2020, doi.org/10.1038/s41598-020-69526-6; Nivard et al., 2024, doi.org/10.1038/s41562-023-01796-2).

Suitably controlling for possible confounding effects of age, sex and environmental factors is a common concern for all genetic association studies, regardless of the phenotype being studied. A standard way to address this in GWAS is to include such factors as covariates in the regression model. Indeed, both Doust et al. (2022) and Niarchou et al. (2022) used age, sex and the top five principal components (estimated from the genetic data of the GWAS participants) to control for potential confounder effects of age, sex and population stratification. They also performed additional relevant follow-up sensitivity analyses. For instance, as noted in our earlier response above, in GWASs targeting different age ranges, Doust et al. (2022) found extremely high genetic correlation (0.97) between older and younger cohorts, indicating little impact of age on the genetic signal that was uncovered. Moreover, for the beat synchronisation GWAS, Niarchou et al. (2022) carried out phenotype validation studies which demonstrated construct validity of the self-reported beat phenotype (in relation to measured beat synchronisation task performance), even after controlling for participants' age, sex, level of education, self-confidence, and musical training. Similarly, potential impacts of sex differences were investigated by Doust et al. (2022) by performing sex-specific GWASs. The genetic correlation between male and female GWASs was 0.91, suggesting little impact of sex on GWAS results. A broader range of environmental variables is not available for the current study, but could be employed in future research to take a more nuanced dive into roles of non-genetic factors.

Epigenetic factors (e.g., histone modifications, chromatin organisation) have potential to impact the aetiology and prevalence of any phenotype, including a wide range of neurological and behavioural traits. Importantly, in the current study we are able to capture the relationship between genetic variation and certain epigenetic imprints by investigating the SNP-heritability enrichment/depletion patterns in genomic annotations that tag epigenetic marks such as promoters and enhancers of various brain cell types, and differentially methylated regions between *Homo sapiens* and archaic human genomes. To that point, the two original GWAS studies also showed enrichment of genetic variants associated with dyslexia and

rhythm traits in foetal and adult brain enhancers and promoters, which are the most important locations in the genome for epigenetic control of brain gene expression. Moreover, here we also integrated our common factor GWAS results with a more detailed set of regulatory element annotations relevant for particular neurobiological and evolutionary processes in the developing and adult brain. These analyses suggest that rhythm- and language-related genetic variation shapes these phenotypes in part by regulating gene expression in the developing and adult brain, with particular links to several brain cell types. With such methods, we can gain understanding of how epigenetic factors may be involved in dyslexia and rhythm impairment, although currently only through investigation of genetic variation.

In addition, following the Reviewer's query about molecular links with songbird vocalisations, we agree that this is a very interesting avenue to explore. We have thus performed a new gene-set enrichment analysis to investigate the links between $F_{\text{gRI-D}}$ and transcriptomic data related to songbird vocal learning. We have added relevant text to the Results and Methods sections, and reported all results in the new Supplementary Table 22:

“We then extended our MAGMA gene-set enrichment analysis to look for potential links between $F_{\text{gRI-D}}$ and genomic substrates of songbird vocal learning, in line with theoretical predictions¹⁰, and prior associations with the genetic architecture of beat synchronisation⁶⁸. To do so, we used nine gene-sets that were curated by Gordon et al. (2021) and therein converted to human homologues for the purposes of gene enrichment analyses; each set represents differential gene expression patterns associated with vocal learning phenotypes (e.g., song vs. silence or number of motifs sung) in Area X and other key regions of zebra finch neural circuitry related to song learning. Intriguingly, we found significant enrichments in four Area X-related gene-sets (Table S22) using the $F_{\text{gRI-D}}$ summary statistics. These findings may suggest overlapping molecular mechanisms between songbird vocal learning, human rhythm, and human language, supporting theories of cross-species convergent evolution of vocal learning and beat perception^{10,69}.” (page 9, lines 252-263)

“In the second part of our MAGMA analysis, we used nine additional songbird-brain expressed gene-sets from Gordon et al. (2021)⁶⁸, and performed a separate gene-set enrichment analysis for $F_{\text{gRI-D}}$. Enrichment p-values were FDR corrected for nine tests.” (page 19, lines 554-557)

We now summarise the above points, and discuss the limitations of our study regarding the residual effects of age, environment, epigenetics, and sex in Discussion:

“Third, even though potential confounders such as age and sex were included as covariates in the dyslexia and rhythm GWAS regression models, we cannot fully exclude residual effects of such factors and other confounders. The original dyslexia GWAS addressed the impacts of age (as noted above) and sex by performing sensitivity analyses using additional age- and sex-specific GWAS, which showed little effect of either of these two factors on GWAS results¹⁸.” (page 13, lines 388-392)

Clarity and context: p.57. rhythm perception has been important already earlier, not only for present day humans. This part of presentation is lacking the evolutionary view to songbirds, monkeys etc.

Musical rhythm skills are indeed not human-specific, and discussing our findings in a broader comparative context is important to have a fuller picture of the evolution of rhythm skills. We now discussed our evolutionary findings considering the songbird and primate rhythm skills literature. to provide a better picture as to how our study contributes to the field:

“Our significant SNP-heritability depletion findings in Neandertal introgressed alleles is in line with findings for other complex human traits⁵⁸, indicating a reduced contribution of Neandertal alleles to reading and rhythm-related skills. Similarly, heritability enrichment findings in primate conserved regions converge with studies that previously found significant enrichments in these loci for complex disease traits⁵⁹. We also noted a trend for the majority of mvGWAS lead SNPs have lower primate conservation scores, indicating weaker constraint on these variants in the primate clade, which may have relevance for the evolution of language and rhythm-related traits on the human lineage. This observation lacks statistical confirmation, but would be in line with prior behavioural and neural findings showing a lack of complex musical rhythm detection and synchronisation in species such as macaque monkeys⁸⁴. Interestingly, one mvGWAS hit locus, mapped to the *DLAT* gene, stood out as strongly conserved among primates, which makes this gene a potential candidate for future experimental investigations in this area. Finally, our genetic signal showed significant enrichments of $F_{\text{GRI-D}}$ -associated

gene sets curated from transcriptome studies of songbird vocal learning, specifically in a key nucleus of the zebra finch brain, Area X. This link between human genetic variants shaping human reading and rhythm skills, and genes involved in songbird song production (e.g., number of motifs sung) in Area X is in line with prior literature⁶⁸, further supporting the importance of shared genetic substrates.” (page 12, lines 341-358)

p. 173-178 the link to many neuropsychiatric diseases mentioned needs justification. Otherwise it looks like they are just cast to the paper.

We thank the Reviewer for pointing this out. To clarify the relevance of this set of analyses to our overall research design, we now changed the first sentence of this paragraph as below:

“To test the validity of links between rhythm impairment and dyslexia risk, and proneness to certain neuropsychiatric disorders proposed by the ARRH, we moved on to investigate relationships of F_{gRI-D} with psychiatric, neurological, and behavioural traits, examining patterns of genetic correlations with common and independent factors in more detail.” (page 7, lines 192-195)

We also interpreted our genetic correlation findings between the common factor and various traits to further highlight the relevance of these results to our work:

“The genome-wide genetic correlation analysis between the common factor and a set of behavioural traits yielded two important findings. First, the genetic correlation directions and magnitudes point to a link between rhythm impairment and dyslexia risks, and neuropsychiatric disorders, in line with the ARRH. Second, language-related traits such as non-word repetition and phoneme awareness yielded the strongest genetic correlations with the common factor of dyslexia and rhythm impairment, further supporting the statistical validity and language/reading relevance of F_{gRI-D} .” (page 11-12, lines 332-338)

**Response to reviewers' comments for Nature Human Behaviour manuscript
NATHUMBEHAV-23103300A titled "The shared genetic architecture and evolution of
human language and musical rhythm"**

- Author responses are in blue. Additions to the manuscript are in blue+bold.
-

Editor Comment: Although the reviewers found your manuscript to have improved during revision, they also raise some important outstanding concerns. We remain interested in the possibility of publishing your study in Nature Human Behaviour, but would like to consider your response to these outstanding concerns in the form of a revised manuscript before we make a decision on publication. In particular, we ask that you fully address the concerns raised by Reviewer #1 surrounding the robustness of the neuroanatomical analyses and the lack of information about the spherical deconvolution tractography method.

We would like to thank the Editor for his guidance with the new set of Reviewer comments. We addressed each Reviewer comment and modified our manuscript based on the concerns raised by Reviewer #1. Specifically, we provided more details on our neuroanatomical analysis method, discussed our SLF-I local genetic correlation finding, generated a new supplementary figure depicting the investigated white-matter tracts in 2D space and made the GWAS summary statistics for the white-matter tracts that are investigated here to allow the reproduction of our local genetic correlation results by independent researchers. Again, we would like to thank all Reviewers for their thorough evaluation of the manuscript, which we believe substantially improved our work.

Reviewer Comments:

Reviewer #1:

Note that line numbers below refer to the clean revised ms, not the version with tracked changes.

The authors have been quite responsive to my comments on the original submission, and the paper is much improved. My comments below are aimed at increasing the clarity and impact of the paper.

We thank the Reviewer #1 for the positive evaluation of our revised manuscript and for additional suggestions. We truly appreciate the thorough and constructive comments in both rounds of revision, which we think greatly improved the manuscript.

Abstract: The final sentence states: “Overall, we provide empirical evidence of multiple aspects of shared biology linking human rhythm and language that could help resolve long-standing debates over their evolutionary relationships.”

If this is to be the “overall” takeaway sentence of the abstract, I suggest changing “human rhythm and language” to “human language and musical rhythm”. Putting language first parallels the paper’s title, and adding “musical” specifies what type of rhythm is being examined, as “human rhythm” is ambiguous since rhythm is present in speech, locomotion, etc.

(Analogously, in line 72, I suggest changing “language development and rhythm” to “language development and musical rhythm”, and in line 295 [first sentence of the Discussion], changing “rhythm” to “musical rhythm”).

We thank the Reviewer for pointing this out. To increase clarity, we rephrased this mention of “rhythm” to “musical rhythm”.

Line 63 refers to “sporadic beat synchronization”, but readers who have not read ref 10 might be confused about what “sporadic” means. I suggest changing this to “predictive and tempo-flexible beat synchronization” (as this mentions key properties of musical beat synchronization) and omitting “sporadic”, which readers can learn about by reading ref 10.

We agree with the Reviewer here. We now replaced the word “sporadic” with “predictive and tempo-flexible beat synchronisation”.

Lines 68-71 state “Some argue that cognitive systems such as language and music are better understood as different uses of overlapping neural mechanisms¹³, yet the genetic and evolutionary bases of putative shared neural circuits are largely unknown.”

The Asano et al. 2022 paper cited in this sentence (ref 13) is agnostic about whether the “overlapping neural mechanisms” occur in overlapping or anatomically segregated neural circuits. That is, I think they argue that language and music can engage similar information processing operations even if those operations occur in adjacent (not anatomically overlapping) circuits. Thus to avoid implying that ref 13 is arguing that language and music use some overlapping neural circuits (which would not be a novel argument – see Patel 2003 Nature Neuroscience), I suggest rewording this sentence to say: “Some argue that cognitive systems such as language and music are better understood as different uses of similar information processing mechanisms¹³, yet the genetic and evolutionary bases of putative shared neural circuits are largely unknown.”

We have reworded this sentence following the Reviewer’s suggestion.

Lines 74-75 say “We believe that identifying potential shared genetic architecture between rhythm- and language-related disorders...” This wording suggests that the paper is examining language disorders in relation to “disorders” of musical rhythm. Calling dyslexia a disorder is already controversial (see Levelt 2001, “Defining dyslexia” *Science*, 292:1300-1301). Are you sure you want to call a problem clapping in time with a musical beat a disorder? This might seem a bit too pathologizing. Perhaps reword this sentence as “We believe that identifying potential shared genetic architecture between language-related disorders and musical rhythm abilities...”

We thank the Reviewer for raising another very important point here. Following this suggestion, we omitted the phrase “rhythm-related disorder”, and reworded this sentence as the Reviewer suggested.

Lines 78-80 say “Importantly, individuals with rhythm impairment have been suggested to show higher predisposition to language-related difficulties such as developmental language disorder and dyslexia (Atypical Rhythm Risk Hypothesis, ARRH)¹⁶.”

Given how central the ARRH is to this paper, it is important that readers know what you mean by “rhythm impairment.” The ARRH paper says “For the purpose of the Atypical Rhythm Risk Hypothesis, we define atypical rhythm as a general construct capturing any/all of the following terms: Impairments in rhythm/beat/meter sensitivity, significantly weaker than normal rhythm ability/skill, poor dynamic attending, beat deafness (Sowinski & Dalla Bella, 2013), or time-based amusia (Peretz & Vuvan, 2017). Atypical rhythm can be classified by poor performance on any implicit or explicit perception or production task of rhythm or timing, such as rhythm discrimination, interval discrimination, rhythm, beat, or meter processing, and synchronization or entrainment.”

It would be worth inserting a small table that summarizes this definition of rhythm impairment, so readers are clear on the phenotypes that the ARRH is concerned with.

We thank the Reviewer for raising another important point. Indeed, rhythm impairment may be a concept that many of the *Nature Human Behaviour* readers are not familiar with. Hence, we added a brief phenotype description box for this term, which we hope clarifies this term:

“Box 1 - Description of the rhythm impairment phenotype:

Rhythm impairment, which is also described as atypical rhythm, refers to impaired (significantly less accurate) performance on a musical rhythm perception or production task (e.g., rhythm and interval discrimination, rhythm or metre processing, beat perception and synchronisation, isochronous motor timing/tapping)¹⁶. It is a broad construct that covers time-based amusia and beat deafness, inaccurate beat

synchronization, related impairments in sensitivity to rhythmic patterns and metre of music, and inconsistent motor timing^{106,107}. The population prevalence of rhythm impairments is estimated to be between 3.0% and 6.5%¹⁷. The Atypical Rhythm Risk Hypothesis (ARRH) claims that rhythm impairment is comorbid with developmental language and speech disorders. While the underlying mechanisms of rhythm impairment are largely unknown, ARRH suggests a shared neurobiological and genetic ground for these traits¹⁶.” (page 22, lines 653-664)

In this revision, the SLF-1 finding is now mentioned in the abstract and is thus highlighted (though oddly, the finding is not mentioned in the Discussion). SLF-1 is not a tract that is typically associated with either language processing or musical rhythm processing, and thus readers familiar with the neurobiology of language or music may treat the finding (and by extension, the entire paper) with skepticism. To protect themselves against such a reaction, the authors should provide more information showing the robustness of their neuroanatomical analyses. Lines 596-600 say: “We extracted the following regions: The left arcuate fasciculus (long, anterior, and posterior segments), the left superior longitudinal fasciculus (I, II, III), and the left uncinate fasciculus for each individual by averaging the FA skeletonised image across a set of five left white-matter tracts defined from a probabilistic atlas¹⁰³.”

It is difficult to assess this part of the analysis because although the authors cite a paper which uses spherical deconvolution tractography, which is presumably what these authors here used to extract the tracts in question, they have not provided enough information to assess the quality of the tractography. It is important to know what software or script they used to run the tractography, with citation; if it was probabilistic or deterministic, and what other parameters were (the number of iterations or streamlines, the curvature setting, whether the tracts were run in native or template space, etc.). The authors also need to indicate what anatomical markers they used, in other words, how they localized and delineated the regions of interest to extract each of the tracts in question. Lastly, and importantly, the tracts in question need to be plotted, either in 2D or 3D space, to be sure they were reconstructed accurately. This plot (along with the other info mentioned above) should be in the supporting information. As a minor comment, I would ask why the authors only reconstructed tracts in the left hemisphere, since the right hemisphere is implicated in auditory processing of non-language sounds which might implicate those areas in beat processing.

As pointed out by the Reviewer, our SLF-I local genetic correlation result was not mentioned in the Discussion, which was a short-coming of our paper. We thank the Reviewer for pointing this out. We now discussed this finding in the Discussion, and suggested potential follow-up analyses to further investigate local and genome-wide genetic correlations between F_{gRI-D} and a refined set of white-matter tracts in future studies:

“Finally, the significant local genetic correlation that we identified between SLF-I and F_{gRI-D} -associated variants in a ~2 megabase region on chromosome 20 represents an interesting example of potential pleiotropic associations between language- and musical rhythm-related traits and white-matter microstructure. This finding is particularly interesting as SLF-I is involved in motor behaviour regulation^{78,79}, suggesting the presence of shared genetic and neuroanatomical elements between motor aspects of language and musical rhythm. It is also plausible that the shared genetic factors underlying language- and musical rhythm-related skills influence a broader range of cognitive processes rather than being confined to the intersection between language and musical rhythm. Here, we note that future local and genome-wide genetic correlation analyses between F_{gRI-D} and a larger selection of neuroanatomical traits (e.g., anterior and posterior segments of the arcuate fasciculus, inferior fronto-occipital fasciculus) and other imaging modalities are necessary to reveal the shared genetics and neuroanatomy of language- and musical rhythm-related traits.” (page 12-13, lines 360-373)

Next, the Reviewer asks why we investigated SLF-I in the first place, pointing out that SLF-I is not a tract typically associated with language or musical rhythm processing. For our local genetic correlation analysis, we adopted literature-informed trait selection criteria and focused our efforts on a small number of white-matter tracts that support various aspects of language- and musical rhythm-related functions. Consequently, we selected five white-matter microstructures that are part of the dorsal or ventral streams of the language network, namely the long segment of the arcuate fasciculus, the superior longitudinal fasciculus subdivisions I-II-III, and the uncinate fasciculus. The reason for limiting our analysis to a small number of tracts was to minimise the multiple testing burden on our local genetic correlation analysis results. Regarding the inclusion of SLF-I subdivision, here are the two main reasons why we think SLF-I is worth looking into in the context of shared genetics of language and musical rhythm:

- 1) SLF-I is part of the dorsal language stream (doi: 10.1016/j.jneuroling.2023.101175), which is a relatively large white-matter bundle with known functions including language production and perception. Even though SLF subdivisions diverge in terms of their anatomic structures, lateralisation patterns, and functions, they develop as adjacent microstructures. Thus, the genetic and molecular programmes shaping SLF subdivisions over the course of embryonic and postnatal neurodevelopment might be shared to a certain extent. As such, better understanding the genetics of SLF-I may provide hints towards understanding the genetics, development and neurobiology of other SLF subdivisions that are adjacent to SLF-I.
- 2) SLF-I connects the superior and medial parietal cortex to the premotor areas, which indicates that it is likely involved in the higher-order and intended movements, as well as the initiation of motor activity (doi: 10.1093/acprof:oso/9780195104233.003.0013). We believe that both of these motor skills may be supporting speech production and rhythmic

entrainment skills, which makes SLF-I an interesting candidate to investigate the shared neurobiology of these two traits.

Overall, our significant local genetic correlation finding between $F_{\text{gRI-D}}$ and SLF-I may suggest that there may be more white-matter microstructures that share their genetic underpinnings with language and musical rhythm skills than currently known tracts.

Following the Reviewer's suggestion, we now elaborated on our neuroanatomical analysis methodology in the Methods section to prevent potential skepticism regarding our SLF-I finding (please see below). The preprocessing of the UK Biobank brain MRI data that we used was performed by the UK Biobank team as described in Alfaro-Almagro et al. (doi: 10.1016/j.neuroimage.2017.10.034). The scripts that were used for preprocessing are publicly available (https://git.fmrib.ox.ac.uk/falmagro/UK_biobank_pipeline_v_1). Specifically, this preprocessing pipeline corrects for eddy current distortions, head motion and outlier-slices (individual slices in the 4D data) using Eddy (<https://fsl.fmrib.ox.ac.uk/fsl/fslwiki/EDDY>). A gradient distortion correction was applied resulting in the 4D output file (see https://biobank.ctsu.ox.ac.uk/crystal/crystal/docs/brain_mri.pdf). Preprocessed images were then fed into the Diffusion Tensor Imaging Fitting software from FSL to generate DTI outputs, including the fractional anisotropy (FA) map. Finally, the DTI FA images were fed into Tract-Based Spatial Statistics analysis. We refrained from reiterating this protocol in our manuscript as this is a pipeline developed and implemented by the UK Biobank team. Secondly, we did not use the imaging-derived phenotypes provided by the UK Biobank. Instead, we averaged the skeletonised images across a set of five standard-space tract masks, which were defined by the Rojkova et al. using the same data processing procedure followed by the UK Biobank and the ENIGMA teams (<http://enigma.ini.usc.edu/protocols/dti-protocols>). Importantly, we did not perform spherical deconvolution tractography, as this was already done by Rojkova et al. (doi: 10.1007/s00429-015-1001-3). Following the Reviewer's suggestion, we now made these points clear in the Methods section. Specifically, we elaborated on the details of our neuroanatomical analysis methodology, corrected the list of five investigated white-matter tracts, and included a new 2D figure depicting the white-matter tracts that are investigated in this study (please see Supplementary Figure 8):

“The diffusion-weighted MRI data were acquired from a 3T Siemens Skyra scanner using the following parameters: isotropic voxel size (resolution)=2×2×2 mm, five non-diffusion-weighted images ($b=0$ s/mm²), diffusion-weighting of $b=1000$ and 2000 s/mm² with 50 directions each, acquisition time is 7 minutes. Whole-brain diffusion-weighted MRI scans were acquired *in vivo*, and fed into the Diffusion Tensor Imaging Fitting toolbox to assess brain microstructure. This analysis created the diffusion tensor imaging (DTI) outputs, including fractional anisotropy (FA) quantitative diffusion maps. Next, the DTI FA images

were fed into the Tract-Based Spatial Statistics analysis¹⁰², resulting in the skeletonised images. Details of the image acquisition, quality control and processing are described elsewhere (refer to https://biobank.ctsu.ox.ac.uk/crystal/crystal/docs/brain_mri.pdf for the full protocol)¹⁰³. We did not use the imaging-derived phenotypes released by the UK Biobank. Instead, we averaged the skeletonised images of five standard-space tract masks defined by Rojkova et al.¹⁰⁴ following a processing protocol similar to the ones applied by the UK Biobank and the ENIGMA teams (<http://enigma.ini.usc.edu/protocols/dti-protocols>). Five left hemisphere white-matter tracts that we investigated here are the long segment of the arcuate fasciculus, the superior longitudinal fasciculus subdivisions I, II and III, and the uncinate fasciculus." (page 21, lines 604-620)

Finally, the Reviewer raises a minor point and asks why we investigated only left-hemispheric tracts. As we have explained above, we focused our efforts on a small set of white-matter structures that are implicated both in language- and musical rhythm-related functions. Given the left hemisphere dominance of the language network in the brain, we narrowed down our tract selection only to left hemispheric microstructures with known speech and musical rhythm associations.

Reviewer #2:

I acknowledge and agree with the author's responses to my comments and revisions to the manuscript. I do not have further queries.

Many thanks to the Reviewer #2 for the positive evaluation of the revised manuscript and constructive comments.

Reviewer #4:

Having read the original submission, the Reviewers 1-3 comments and authors' answers, plus the revised version of the manuscript, I'm quite convinced of the work in total. The issues raised, for example concerning the validity of the phenotypes such as self-assessed ability to beat to the rhythm and self-reported dyslexia have been well taken into consideration and explained to the reader. Personally, I quite like the idea of utilizing such mass data as collected by eg companies like 23andMe, with self-assessed traits, cause compiling this large numbers of individuals into a study is by other means still quite unaccomplishable or extremely costly. Such data should definitely be utilized as far as possible.

Methodologically, the work is outstanding, with a variety of newest analytical techniques applied. I especially appreciate the methods taking the different evolutionary time frames into consideration, where the natural selection might have taken place in different time depths of primate and human evolution.

The revised manuscript handles well the points made by the reviewers. I warmly recommend the manuscript to be accepted to be published by the Nature Human Behaviour.

We would like to thank Reviewer #4 for the very positive evaluation of our manuscript and our previous responses to the Reviewers.